# Classification Of Large-Scale Environments That Drive The Formation Of Mesoscale Convective Systems Over Southern West Africa

Francis Nkrumah[1,2], Cornelia Klein[3,5], Kwesi Akumenyi Quagraine[1,4], Rebecca Berkoh Oforiwaa[2,6], Nana Ama Browne Klutse[2,6], Patrick Essien[1,2], Gandomè Mayeul Leger Davy Quenum[2,7] and Hubert Azoda Koffi[6]

[1]Department of Physics, University of Cape Coast, Private Mail Bag, Cape Coast, Ghana;
[2]African Institute of Mathematical Sciences (AIMS), Sector Remera, Kigali 20093, Rwanda;
[3]U.K. Centre for Ecology and Hydrology, Wallingford, United Kingdom
[4]Climate System Analysis Group (CSAG), ENGEO, University of Cape Town, Private Bag X3, Rondebosch, Cape Town 7701, South Africa
[5]Department of Atmospheric and Cryospheric Sciences, University of Innsbruck, Innsbruck, Austria
[6]Department of Physics, University of Ghana, Legon P.O. Box LG 63, Ghana
[7]National Institute of Water (NIW), University of Abomey-Calavi, Godomey, Cotonou 01 PB: 4521, Benin

*Correspondence to*: Francis Nkrumah (francis.nkrumah@ucc.edu.gh) and Nana Ama Browne Klutse (nklutse@ug.edu.gh)

**Abstract.** Mesoscale convective systems (MCSs) are frequently observed over southern West Africa (SWA) throughout most of the year. These MCS events are the dominating rain-bearing systems, contributing over 50% of annual rainfall over SWA. However, it has not yet been identified what variations in typical large-scale environments of the West African monsoon seasonal cycle may favour MCS occurrence in this region. Here, nine distinct synoptic states are identified and are further associated with being either a dry, transition, or monsoon season synoptic circulation type using self organizing maps (SOMs) with inputs from reanalysis data. We identified a pronounced annual cycle of MCS numbers with frequency peaks in April and October that can be associated with the start of rainfall during the major rainy season and the maximum rainfall for the minor rainy season across SWA respectively. Comparing daily MCS frequencies, MCSs are most likely to develop during transition conditions featuring a northward-displaced moisture anomaly (2.8 MCSs per day), which can be linked to strengthened low-level westerlies. Considering that these transition conditions occur predominantly during the pre- and post-monsoon season, these patterns may in some cases be representative of monsoon onset conditions or a delayed monsoon retreat. On the other hand, under monsoon conditions, we observe weakened low-level south-westerlies during MCS days, which reduce moisture content over the Sahel but introduce more moisture over the coast. Finally, we find a majority of MCS-day synoptic states to exhibit positive zonal wind shear anomalies. Seasons with the strongest zonal wind shear anomalies are associated with the strongest low-level temperature anomalies to the north of SWA, highlighting that a warmer Sahel can promote MCS-favourable conditions in SWA. Overall, the SOM-identified synoptic states converge towards high moisture and high shear conditions on MCS days in SWA, where the frequency at which these conditions occur depends on the synoptic state.

## 1 Introduction

The region of West Africa is subject to variability in rainfall on both spatial and temporal scales. Fundamentally, the rainfall pattern in West Africa is modulated by the annual change in the position of the Intertropical Convergence Zone (ITCZ) and the West African Monsoon (WAM). Due to endemic poverty, lack of infrastructure and technology, rapid population increase, and significant fluctuation of the WAM, West Africa has been deemed one of the world's most susceptible regions to climate change (IPCC, 2014). The climate of southern West Africa (SWA) can be categorized into four seasonal stages: a dry season from December to February, two wet seasons lasting from April to June, and September to November, and the so-called little dry season in August (e.g. Thorncroft et al. 2011). Between March and June, when low-level winds are more westerly and the intertropical convergence zone (ITCZ) starts to move northward, the precipitable water peaks over SWA (Klein et al. 2021). The ITCZ retreats southward in September, creating the second rainy season, followed by a dry season from November to January.

One major atmospheric disturbance that contributes to the WAM is the presence of Mesoscale Convective Systems (MCSs) which supplies around 30-80 % of the total rainfall during the WAM (Klein et al. 2018). MCSs are organized thunderstorm clusters, often defined to have a minimum horizontal extent of the precipitating area of 100 kilometres in at least one direction (Guo et al. (2022); Chen et al. (2022); Houze (2004)). Maranan et al. (2018) note that diverse MCS sub-groups such as squall- or disturbance lines, structured convective systems, and mesoscale

convective complexes impact the hydro-climate of West Africa. In both the tropics and midlatitudes, MCS also contributes significantly to rainfall extremes, rendering them a substantial contributor to the hydrologic cycle (Feng et al. (2021); Li et al. (2020)). More studies have been motivated in recent decades by evaluating drivers that affect rainfall variability and intensity associated with MCSs (Baidu et al. (2022); Augustin et al. (2022)). MCSs, for instance, supply essential precipitation and, as a result,  supply water to agriculturally productive regions in the tropics, particularly in semi-arid regions such as the Sahel (Nesbitt et al. (2006)).

However, relative to our understanding of MCS drivers in the Sahel, SWA has received less attention. The connections of MCSs to larger-scale atmospheric motion and states are both important and not fully understood for the southern region, hence, a better understanding of large-scale MCS drivers is important for improving precipitation prediction over SWA. Earlier research has suggested an increasing role of other types of less-organized rainfall in place of MCSs over the Guinea Coast (e.g. (Acheampong, 1982; Fink et al., 2006; Kamara, 1986; Omotosho, 1985), with MCS contribution to annual rainfall decreasing from 71% in the Soudanian to 56% in the coastal zone (Maranan et al 2018), emphasizing MCS importance across the SWA region. Maranan et al., 2018 also concluded that precipitable water and Convective Available Potential Energy (CAPE) determine where MCSs may occur in SWA, while zonal wind shear is a stronger predictor for distinguishing between small scattered convection and MCS-type development. Indeed, zonal wind shear intensification was found to be a major driver of increasing frequencies of the most intense Sahelian MCSs over the last three decades (Taylor et al., 2017), a mechanism that was similarly found to play a role for early-season MCS intensification in SWA (Klein et al 2021). Zonal wind shear, which is thought to modulate the storm-available supply of moist buoyant air, is also seen to be very critical to the organization of convective systems (e.g., Alfaro, 2017; Mohr & Thorncroft, 2006). Accordingly, propagating storms with longer-lasting organized precipitation systems were consistently found to be associated with strong vertical wind shear and higher values of CAPE in the Sahel (Hodges & Thorncroft, 1997; Laing et al., 2008; Mohr & Thorncroft, 2006).

Previous studies address the large-scale settings for WAM-related rainfall throughout the seasons (Sultan and Janicot, 2003) with less attention given to the importance of large-scale WAM modes and their effect on regional MCS frequencies in SWA. The role of regional MCS-centred environments in the initiation and development of MCSs in West Africa has been well studied (e.g., Klein et al. 2021; Vizy and Cook 2018; Schrage et al. 2006; Maranan et al. 2018). Vizy and Cook (2018) observed that the extension of vertical mixing to the level of free convection, as a result of surface heating, tends to initiate MCSs in an environment where the mid-tropospheric African easterly wave disturbance is located in the east. The vertical wind shear is enhanced as a result of the synoptic disturbance. Klein et al. (2021) suggested that heavy rainfall, due to cold MCSs during both dry and rainy seasons, occurs in an environment with stronger vertical wind shear, increased low-level humidity, and drier mid-levels. Unlike vertical wind shear, Maranan et al., (2018) suggested that thermodynamic conditions such as CAPE and Convective Inhibition (CIN)  are of lesser importance for the horizontal growth of convective systems, although they indicate the potential of the initial vertical development of convective systems. Janiga and Thorncroft (2016) also suggested that CAPE, vertical wind shear and column relative humidity are the decisive large-scale environmental parameters that control the characteristics of convective systems. Based on radar and sounding

observations aligned around 15ºN, Guy et al. (2011) analyzed MCSs and their respective environmental conditions
over three different regimes of West Africa (maritime, coastal, and continental). They concluded that MCSs tend to
occur ahead of the African easterly wave (AEW) trough during the maritime and the continental regime, while they
are mostly found behind the trough in the coastal regime.

It is not clear to what extent different large-scale patterns of atmospheric drivers such as temperature, wind,
humidity, and CAPE at different stages of the WAM drive the formation of MCSs over SWA. The SWA region
differs from its Sahelian counterpart in its closer proximity to the ocean and a distinct bimodal rainfall seasonality.
The WAM stages can broadly be classified into a dry season when north-easterly Harmattan winds prevail over most
of West Africa during December-February when rainfall mostly occurs off the southern coast of the continent
(Thorncroft et al 2011), and the monsoon season from July-September, initiated by a striking jump of the monsoonal
rainfall band from coastal regions to the Sahel (Hagos and Cook, 2007). The monsoon months thus represent the
unimodal Sahelian rainfall season. In SWA however, the majority of rainfall occurs between the dry months and
monsoon months, when the monsoon rainband first passes northward over southern regions from March to June, and
subsequently moves southward again when the monsoon retreats in October (e.g. Maranan et al 2018, Klein et al
2021). Here, we define these months when SWA receives most of its rainfall as transition season.
From this SWA perspective, our study systematically classifies the different large-scale patterns across the WAM
region and how they are associated with MCSs over SWA. For this purpose, a classification using a self organizing
map (SOM; Kohonen 2001) analysis was carried out to characterize large-scale WAM patterns during the 1981-
2020 period, which we subsequently grouped into days with MCS occurrence over SWA. The SOM is a clustering
technique that is topologically sensitive and uses an unsupervised training method to cluster the training data
(Lennard and Hegerl, 2014; Quagraine et al. 2019). This methodology thus allows us to identify favourable types of
large-scale environments driving the formation of MCSs within different WAM stages.
The paper is organized as follows: Section 2 details the study area and data sources and how they were
processed. In section 3, the SOM methodology and other needed statistics used to investigate the relationship
between large-scale environment patterns and particular MCSs are presented. Section 4 discusses the main results,
which include the common features and different types of large-scale patterns associated with MCSs. Section 5
provides the summarized conclusions of the study.
**2 Data Sources and Processes**
**2.1 ERA5 Reanalysis Data and MCS Data**
The ECMWF fifth-generation atmospheric reanalysis (Hersbach et al., 2020), ERA5, was used as the main
data source in this work. The dataset is generated using 41r2 of the Integrated Forecast System (IFS) model, based
on a four-dimensional variational data assimilation scheme, and takes advantage of 137 vertical model levels and a
horizontal resolution of 0.28125º (31 km). The data provides hourly estimates of model integration. In this study,
hourly zonal and meridional winds (650 and 925 hPa), specific humidity (925 hPa), temperature (925 hPa), and
convective available potential energy (CAPE) in ERA5 during 1981–2020 were used to explore suitable large-scale
environments for the development of MCSs in SWA (5-9°N, 10°W-10°E). The zonal and meridional wind at 925
hPa, are used to understand the penetration of monsoon flow inland. The zonal wind difference between 925 hPa
and 650 hPa is used as a zonal wind shear change indicator while the temperature at 925 hPa is used to visualize
Saharan heat low (SHL) differences. Due to the main direction in which MCSs propagate (east to west), enhanced
easterly zonal wind shear are presented as positive anomalies as these are positively related to storm development.
Specific humidity (q) at 925 hPa was used to explore whether CAPE changes are controlled by low-level q. We
consider also the total column water vapour (TCWV) due to its ability to represent the total gaseous water in the
vertical column of the atmosphere which is influenced by the evolution of the humidity field. The Meteosat Second
Generation (MSG) cloud-top temperature data, which are available every 15 minutes from the Eumetsat archives
online (https://navigator.eumetsat.int/product/EO:EUM:DAT:MSG:HRSEVIRI) was used in this study. Twelve
years of MCS snapshots (2004–15) detected from Meteosat Second Generation 10.8 µm-band brightness
temperatures (Schmetz et al., 2002, EUMETSAT 2021) are used to define MCS days in this study. Following (Klein
et al., 2021), an MCS is defined here as a -50°C contiguous cloud area larger than 5000 km$^2$. We consider the MCS
images every half hour, for which they are matched up with the half-hourly Integrated Multi-satellite Retrievals for
Global Precipitation Measurement (IMERG; Huffman et al. 2019) dataset, using the merged microwave/infra-red
("precipitationCal") rainfall product. An "MCS day" is then defined as a day with at least one hour containing 5
simultaneously existing MCSs between 16 and 1900 UTC with maximum rainfall >5mm within the SWA domain.
Here, only land-based MCSs are considered because MCSs over land are fundamentally more intense and deep than
its counterpart over the ocean (Mohr and Zipser 1996).

**3 Methodology**
**3.1 Self-organising Maps (SOMs) analysis**
The study uses the self organizing map (SOM; Kohonen 1982, 2001) from SOM-PAK-3.1 software. The
technique is used to identify archetype synoptic circulation patterns over the southern West Africa region by training
a 9-node SOM with ERA5 daily mean 925 hPa geopotential height fields to produce 9 characteristic circulation
patterns for the period 1981 to 2020. The geopotential height circulation pattern is used here mainly based on its
physically realistic output spanning a range of circulation features found in the atmosphere (Hewitson and Crane,
2002) and its ability to detect the West African Heat Low (WAHL) which is a key element of the West African
monsoon system (Lavaysse et al. 2009; Biasutti et al. 2009). The SOM is mostly the preferred choice over other
clustering methods such as the principal component analysis (PCA) or K-means because the data is not discretized
and orthogonality is not forced or does not require subjective rotations to produce interpretable patterns. The main
advantage of the SOM technique is its ability to deal with non-linear data (such as the continuum of atmospheric
conditions) and can easily be visualized and interpreted (Reusch et al. 2005; Lennard and Hegerl, 2014). The steps
within the technique can be broadly grouped into two stages, namely the training stage and the mapping stage.
Earlier studies (e.g. Hewitson and Crane 2002; Kim and Seo 2016; Lee 2017; Rousi et al. 2015; Sheridan and Lee
2012) have successfully used this technique in synoptic climatology to effectively preserve relationships between

weather states while giving outputs that are readily understood and can be easily visualized as an array of classified patterns. These classified patterns help in interpreting relationships between large-scale regional circulation patterns and local weather expressions and rainfall extremes (Hewitson and Crane 1996; Cassano et al. 2015; Wolski et al. 2018). In this study, the SOM is randomly initialized allowing for hidden patterns and structure in the geopotential height at 925 hPa to be discovered while the algorithm iteratively updates the weights of the nodes to better represent the data. The strength of initializing the SOM this way lies also on its robustness to noise and outliers as a result of the algorithm applying a competitive learning structure to the data which then allows for the formation of distinct clusters. The SOM_PAK algorithm allows the SOM process to minimize quantization and topological errors at the mapping stage when choosing the best SOM as outlined in Lennard and Hegerl (2014). However, there is a trade-off when choosing the size of the SOM, as this is dependent on the need to generalize circulation states for analysis or the need to capture predominant spatial characteristics that affect the local climate. The choice of how many SOM nodes is a trade-off between distinctiveness and robustness. Based on SOM_PAK, we tested node sizes 2x3, 3x3, and 3x4, using the quantization error (QE) as an indicator of the quality and robustness of the respective node size. We find a minimized QE for 3x3 (c.f. Supplementary Figure S1), which, from visual inspection, also shows a larger number of distinct circulation features than 2x3 while producing fewer redundancies than 3x4. Thus, all the following analyses are based on the 3x3 node matrix.

**3.2 Large-scale WAM patterns on southern West Africa MCS days**

Based on the 9 different large-scale node patterns, we explore within-node large-scale conditions that characterize MCS days in SWA. For examination of environmental conditions suitable for SWA MCS activity, large-scale conditions were taken from hourly ERA5 reanalysis data sampled at 1200 UTC when the daily convective activity is more representative of pre-convective atmospheric conditions (Klein et al. 2021). Pre-convective conditions are considered in the study to reduce the effects of feedback from the MCSs on environmental conditions (Song et al. 2019). Composites of ERA5 large-scale environmental variables (temperature, wind, specific humidity, and CAPE) are created for all node days, and for MCS days within each SOM node. Finally, the anomaly in large-scale patterns between MCS days and node mean conditions are computed to determine MCS-favourable adjustments in large-scale patterns within each node. A two-sided Student's t-test is used to determine significant differences between node climatologies and MCS-day sub-samples.

In addition to large-scale condition composites, we also sample pre-convective (1200 UTC) local atmospheric conditions (ERA5), for each 1800 UTC MCS at the location of minimum cloud top temperature. We only consider 1800 UTC MCSs for local condition sampling to avoid oversampling similar atmospheric states from several MCS time steps. These conditions are compared to the node climatology conditions at the same locations, allowing us to explore the difference in node climatology versus MCS day conditions at the specific locations where MCSs occurred on respective days. Here we only focus on the afternoon peak of convection when it is triggered and is in early stages of organization. It should be noted that driver importance may shift for nocturnal MCSs in later hours, when CAPE is reduced over night and shear may increase further in importance for MCS maintenance (Vizy et al, 2018)

## 4 Results

### 4.1 Node seasonality and mean conditions

A 9-node SOM (Fig. 1) with distinct synoptic states was identified, where the nodes are hereafter referred to as nodes one (1) to nine (9). Considering the SOM node frequency distributions in Fig. 1, it is noticeable that the nodes separate different stages of the monsoon circulation seasonality, although certain nodes evidently cover a wider range of months that cannot be represented by the typical monthly grouping of the seasonal cycle (e.g. 2,3,5,8). Circulation patterns in nodes 1, 4, and 7 can be attributed to cases primarily observed in the first three months (January, February, and March) and the last two months (November, and December), hence a pattern most representative of the dry season months. On the other hand, nodes 2, 5, and 8 depict an environment that is prominent during the pre-monsoon and the post-monsoon seasons, with node 2 presenting a clearer seasonal exclusivity during pre-monsoon while nodes 5 and 8 show frequent occurrences during the post-monsoon season. These nodes (nodes 2, 5, and 8) are hence in the following referred to as transition season nodes, a period that connects the dry and monsoon season. The right-hand side of the SOM nodes 3, 6, and 9 represent patterns that cover monsoon season months, but can similarly feature high frequencies outside of the monsoon season (e.g. node 3 with the highest frequency in May).

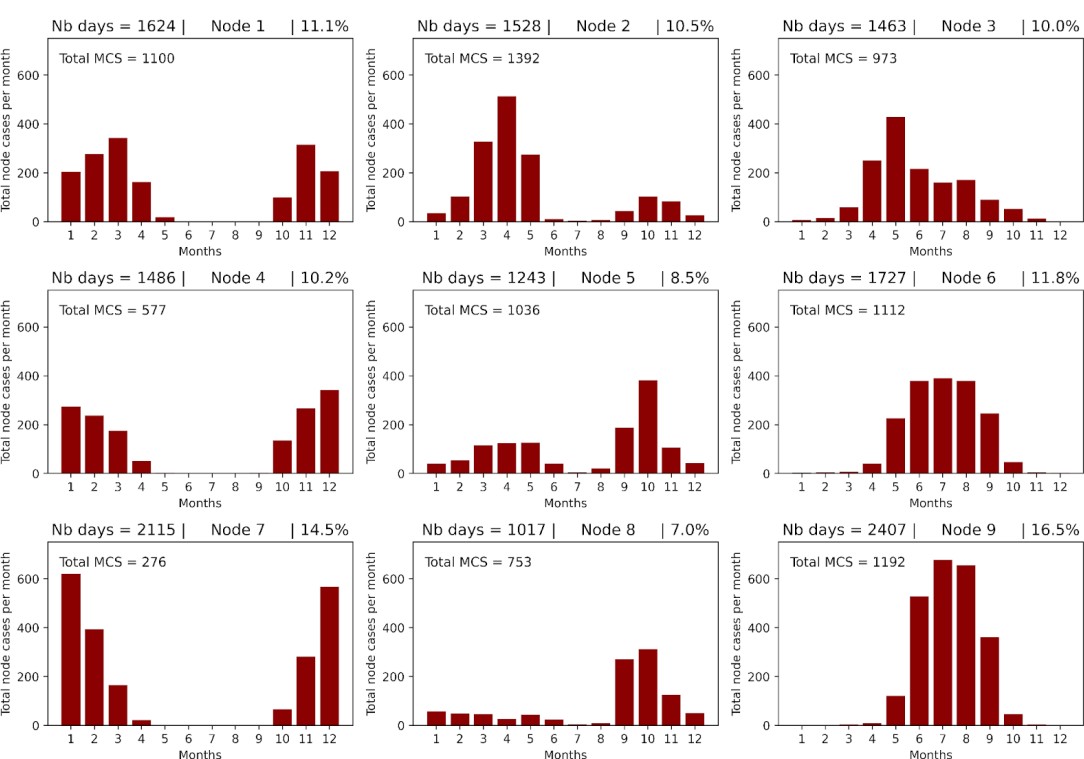

**Figure 1.** Monthly distribution of node cases based on SOM analysis. Bar values indicate the total number of MCSs
per month from 2004 to 2015. The total number of MCS per node from 2004 to 2015 is displayed in node panels.
The title shows the total number of days in each node (left) and the contribution of each node to the total node days
(right).


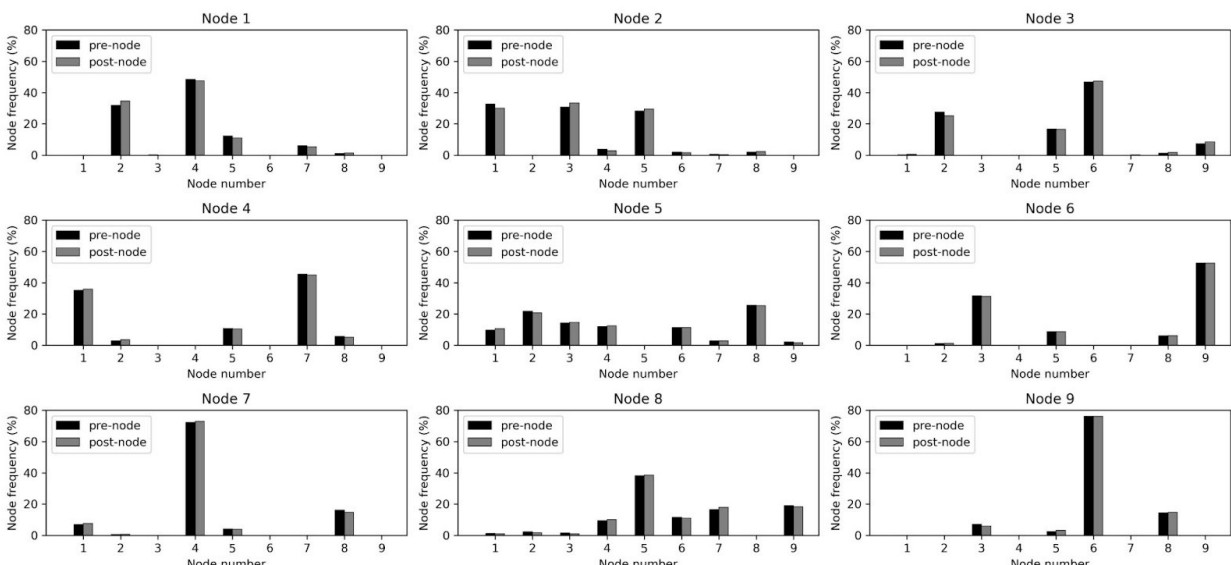


**Figure 2:** Frequency of nodes (%) preceding (pre-node) or following (post-node) each of the nine nodes.

To investigate the relationship between nodes across our 3x3 SOM matrix, we now consider the

frequencies at which node states are preceded or followed by other nodes in Fig. 2. The resulting frequency
distributions reinforce a classification of the matrix columns into dry (1,4,7), transition (2,5,8) and monsoon (3,6,9)
season nodes, with the top row (1,2,3) representing nodes that are preceded or followed by nodes of a different
season (column) 20-30% of the time. The bottom row nodes (7,8,9) on the other hand are distinct within-season
states that are almost never connected to first row nodes (1,2,3) but are reached via intermediate middle row nodes
4,5,6. The node matrix separates different season states along rows, while columns seem to represent within-season
states where upper and lower rows are separate states, temporally connected by conditions captured by middle-row
nodes. Finally, the persistence of nodes presented in Fig. 3 reflects the discussed matrix structure, with connecting
middle-row nodes 4,5,6 featuring shortest periods with on average 1.7-1.9 days, suggesting more transient states.
Nodes 2,9,7 on the other hand show the smallest number of single day occurrences (consecutive node days = 1),
pointing towards more stable, persistent conditions with an average period length of 3.8-4.3 days. Regarding node
characteristics, it is striking that each seasonal node group features nodes of differing persistence (c.f. node season
order for consecutive node days = 1), rendering node persistence a key difference between same-season nodes in the
SOM matrix columns.

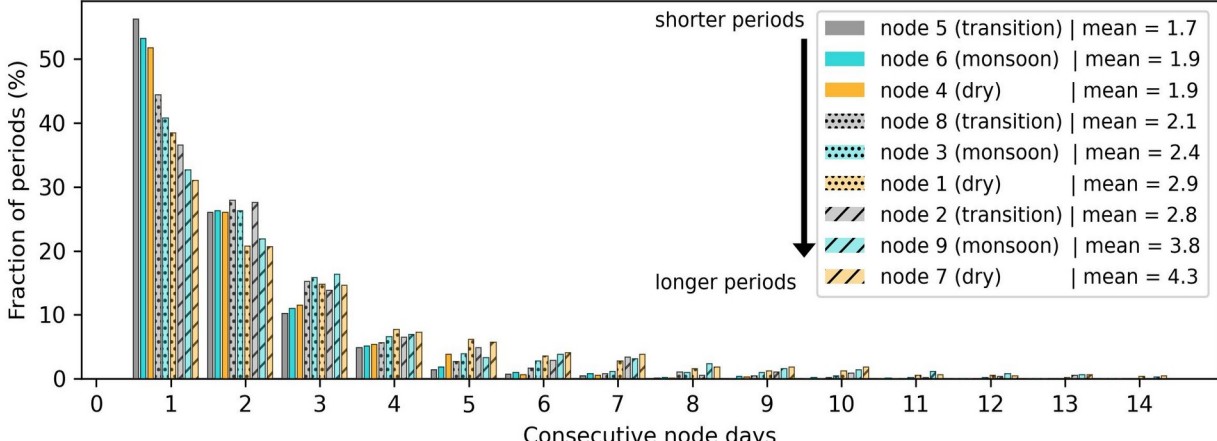


**Figure 3:** Fraction of periods covering consecutive days of different lengths per node, with the total percentage for
1-14 consecutive node days adding up to 100% per node. The node bars are ordered according to the period fraction
for "consecutive node days = 1", revealing the node order going from shorter to longer temporal node persistence, as
shown in the legend.

In the following, we inspect the average atmospheric conditions associated with the identified nodes. The
SOM classification of different synoptic states was based on 925 hPa geopotential heights, with resulting patterns
shown in Fig. 4. The patterns clearly show the signature of the well-known West African Heat Low (e.g. Lavaysse et
al. 2009) moving northwards, strengthening over the course of the annual WAM cycle (from nodes 1, 2, and 3) and
peaking in August, evident as an area of low pressure over the Sahara in nodes 3, 6, and 9. Nodes 4, 7, and 8 show
stages of the weakening of the heat low, coinciding with a southward movement of the 925 hPa low pressure area.
The overlaid 650 hPa wind field reveals mean easterly wind conditions at MCS steering levels across all nodes,
suggesting that the dominant propagation direction for MCSs remains east to west for all identified synoptic states.
As was shown in Fig. 3, the discussed node states have an average duration on the order of days, indicating frequent
transitions. Notably, mid-level westerlies are strengthened or shifted southwards for all top-row nodes in  Fig. 4,
which is associated with increased probability for  MCS occurrence compared to other nodes, as we will outline later
(c.f. Fig. 8). Potential synoptic factors that may drive the frequent node transitions and hence affect MCS frequency
include extratropical waves, as well as the WAHL that is most pronounced for top-row nodes. WAHL variations
were shown to take place on the order of days, in some cases modified by dust concentration (Lavaysse et al. 2011),
while its southward expansion on sub-seasonal timescale has been associated with higher shear and more intense
MCSs in SWA (e.g. Talib et al. 2022).

9    9


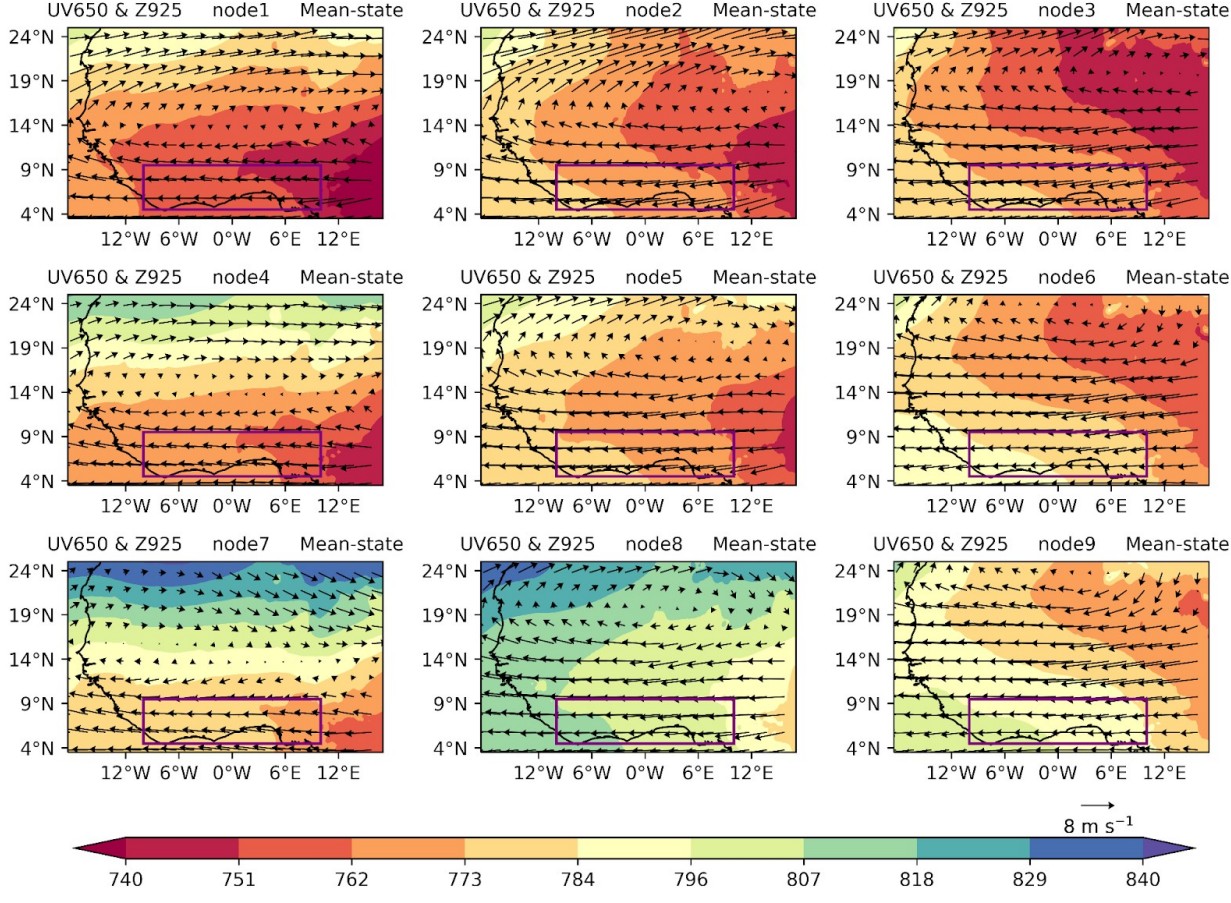


**Figure 4.** 12 UTC composites of 925-hPa geopotential height (shading; gpm) and 650-hPa winds (vectors; m s$^{-1}$ ) in
9 nodes based on SOM analysis. The purple box depicts the SWA region (5º–9ºN, 10ºW–10ºE)

We now examine winds and moisture flows at 925 hPa to explore their behaviour under the nine distinct
circulation types identified (Fig. 5). In nodes 1, 4, and 7, the north-easterly winds dominate most of West Africa,
with weak southerlies over SWA. This pattern in moisture distribution is evident in the dry season over West Africa,
signaling a low moisture presence. The enhanced moisture observed in coastal areas of SWA can be attributed to the
penetration of southerly winds. In the transition node 2, the southerly winds strengthen and move inland, causing the
north-easterly winds to retreat. A similar effect is observed in nodes 5 and 8 where the north-easterlies become
weaker. In nodes 3, 6, and 9, the south-westerlies are intensified and move inland, further enhancing moisture flow
from the South Atlantic towards the land, representative of monsoon flow. Wind patterns for mid- and low-levels
(Figs. 4 and 5) illustrate vertically-sheared conditions coinciding with regions of high low-level specific humidity in
all nodes (purple in Fig. 5), thus marking regions where atmospheric conditions may allow MCS development.

10     10

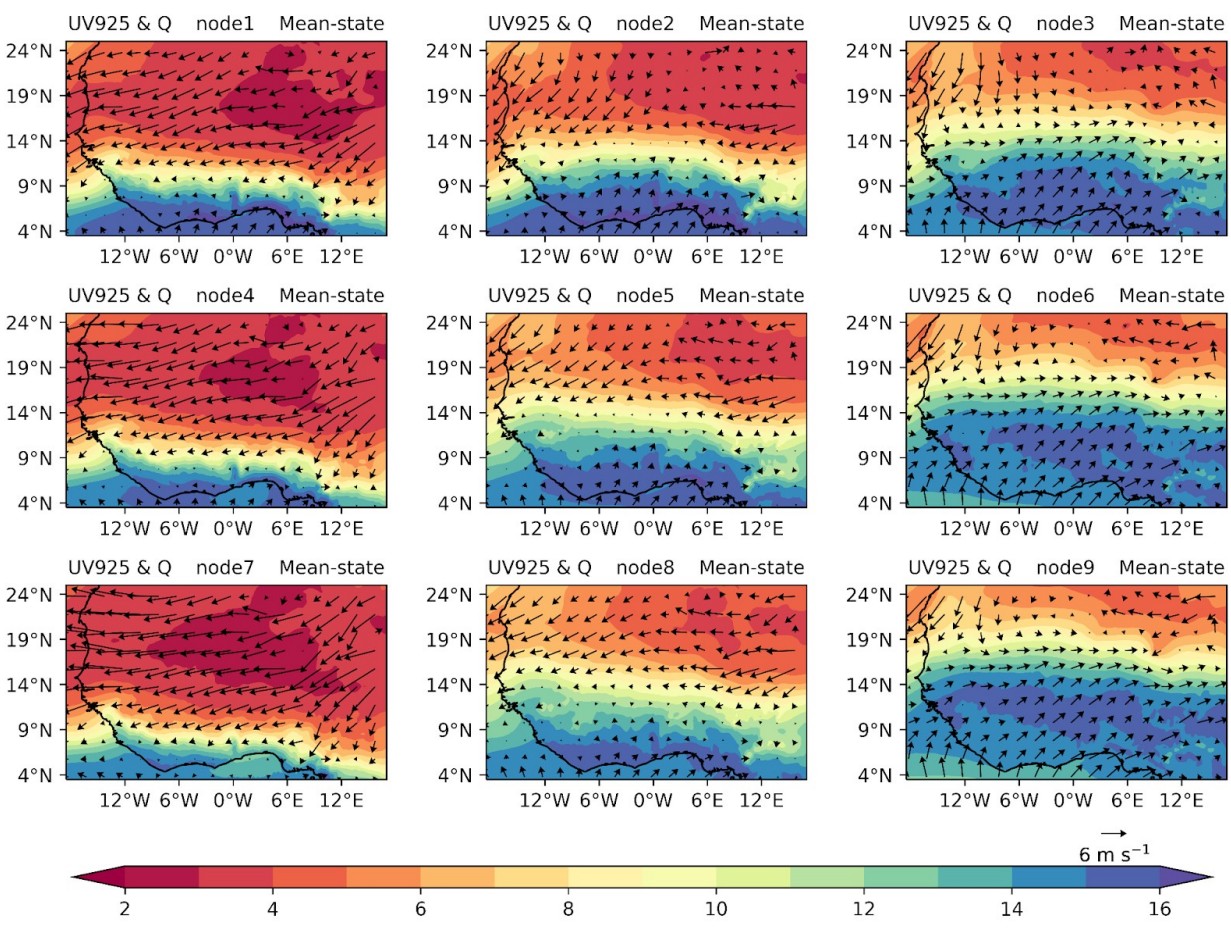

**Figure 5.** 12 UTC composites of specific humidity (shading; g kg$^{-1}$) and 925-hPa winds (vectors; m s$^{-1}$) in 9 nodes based on SOM analysis.

A further investigation was conducted to ascertain the spatial distribution of mean zonal wind shear over SWA (Fig. 6), where easterly shear is represented with a positive sign in this study as it is easterly shear that contributes to MCS development in this region. The patterns in zonal wind shear demonstrate northward transport during the propagation of the WAM cycle and a wider spread of zonal wind shear from first to third column nodes illustrate a strong link of high-shear areas to the propagation of the WAM cycle, and these areas widen as the zonal shear band moves further inland. High-shear areas also closely follow the northern boundary of increased low-level humidity, marking the areas where humidity and shear conditions may allow MCS development. For nodes with high frequency in the monsoon season (nodes 6 and 9), zonal wind shear peaks clearly to the north of the SWA domain. A southward retreat of zonal wind shear is observed during the post-monsoon season (nodes 2, 5, and 8).

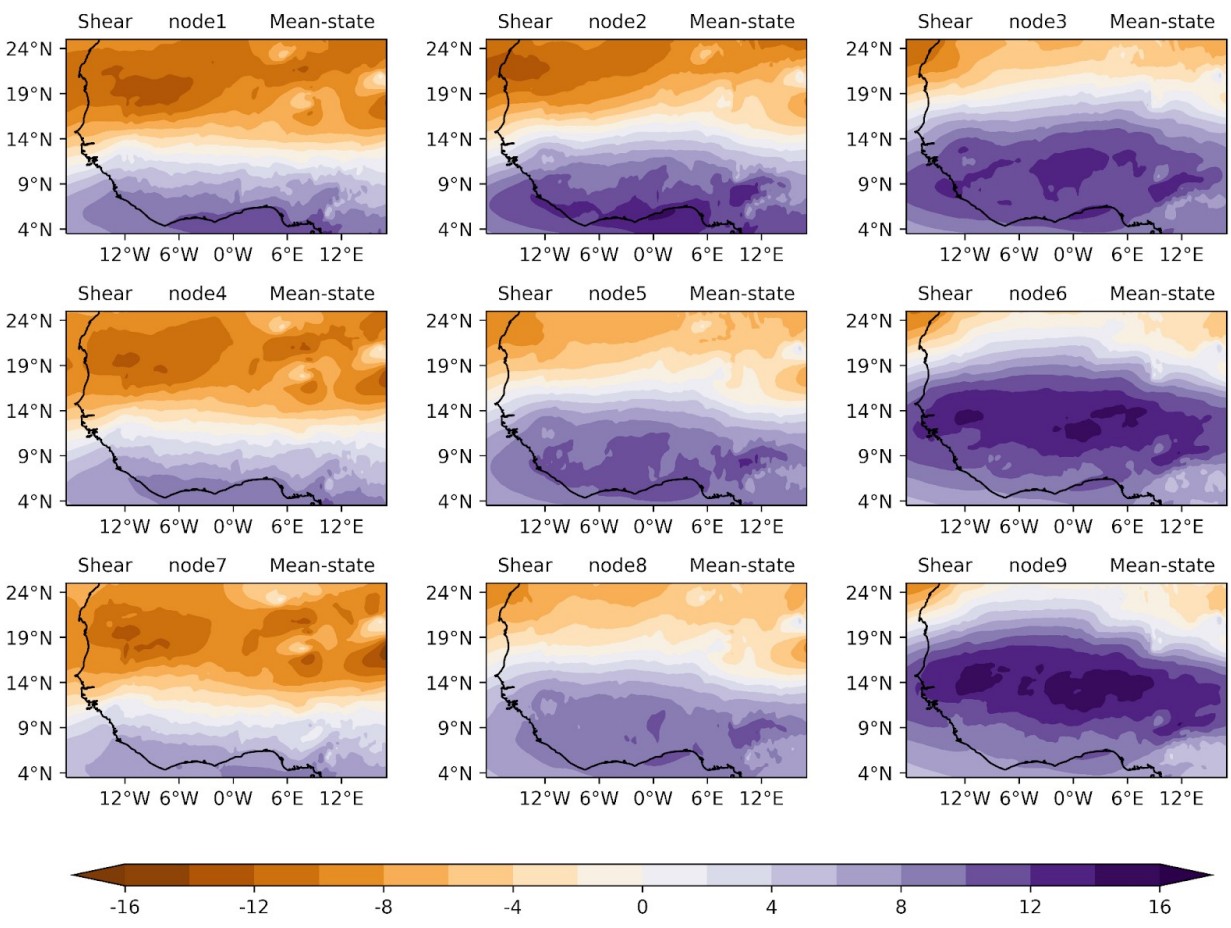

**Figure 6.** 12 UTC composites of zonal wind shear in 9 nodes based on SOM analysis.

## 4.2 Large-scale conditions favouring MCS days

The environmental conditions that are associated with MCS occurrence are described in this section. Firstly, the monthly climatology of MCS frequency as captured by our MCS snapshots (average number of MCSs at 1800 UTC across SWA domain) is considered with a focus on rainfall months in Fig. 7, which shows a pronounced annual cycle of MCS numbers with frequency peaks in April and October. These peak months are associated with the start of rainfall during the major rainy season and the maximum rainfall for the minor rainy season across SWA respectively. The monthly climatology of MCS frequency decreases from April to August, with August being the local minimum. This local minimum corresponds to the so-called "little dry season" (Le Barbé et al., 2002; Vollmert et al., 2003) that exists before the southward retreat of the rainbelt.

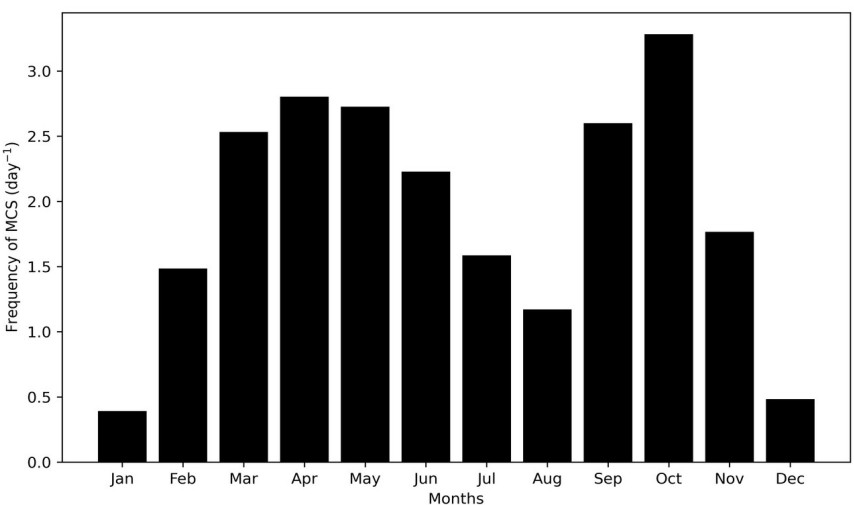

 **Figure 7.** Average annual cycle of MCSs at 1800 UTC within the SWA box showing the monthly average of MCS

number per day.

The spatial distribution of MCS frequencies during node days is depicted in Fig. 8. Comparing daily MCS
frequencies, we find that MCSs are most likely to develop under transition node (2,5,8) conditions (2.8 MCSs per
day) featuring a northward-displaced moisture anomaly (Fig. 9). Given the transition nodes occur predominantly
during pre-monsoon (late March to June) and post-monsoon (from September to November) - the major and the
minor rainy season respectively in SWA (cf. Fig.~1), these patterns may in some cases be representative of early
monsoon onset and a delayed monsoon retreat respectively. MCSs rarely develop under dry node (1,4,7) conditions,
with frequencies as low as 0.6 MCSs per day. Frequency signals in node 1 are dominated by land-sea breeze
convection along the coast which are gradually suppressed in nodes 4 and 7. Large-scale settings, therefore,
seemingly facilitate such rather local-scale developments. Nodes 1 and 9 feature the same overall MCS frequency,
where node 1 however shows coastal MCS frequency peaks as is representative for dry season characteristics, while
MCS frequency peaks are shifted towards the Sahel during node 9 monsoon conditions.


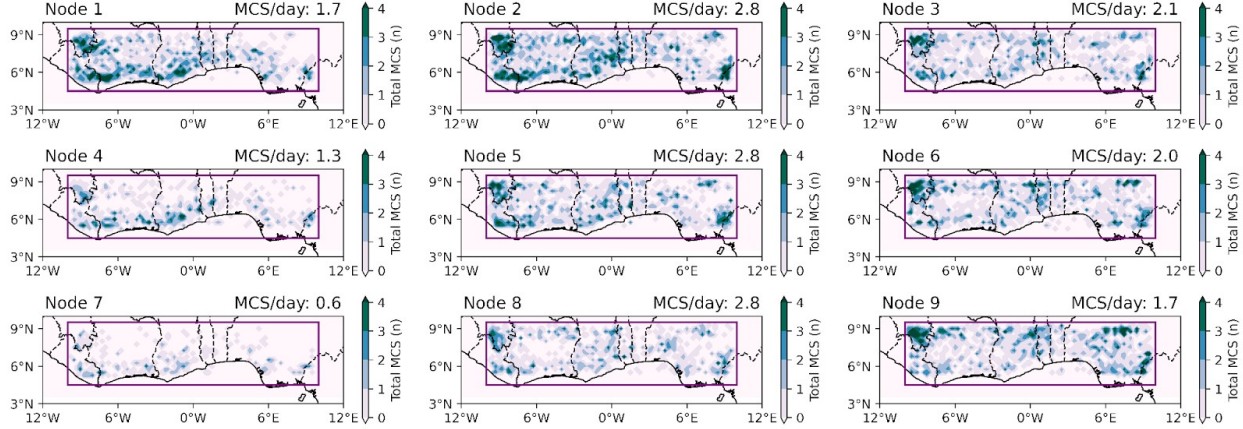


**Figure 8.** The SWA region indicating the spatial distribution of MCSs during node days. The purple box depicts the
main study region of southern West Africa (SWA, 10ºW - 10ºE, 5-9ºN) and titles show the frequency of MCS per
day per node within the SWA box.


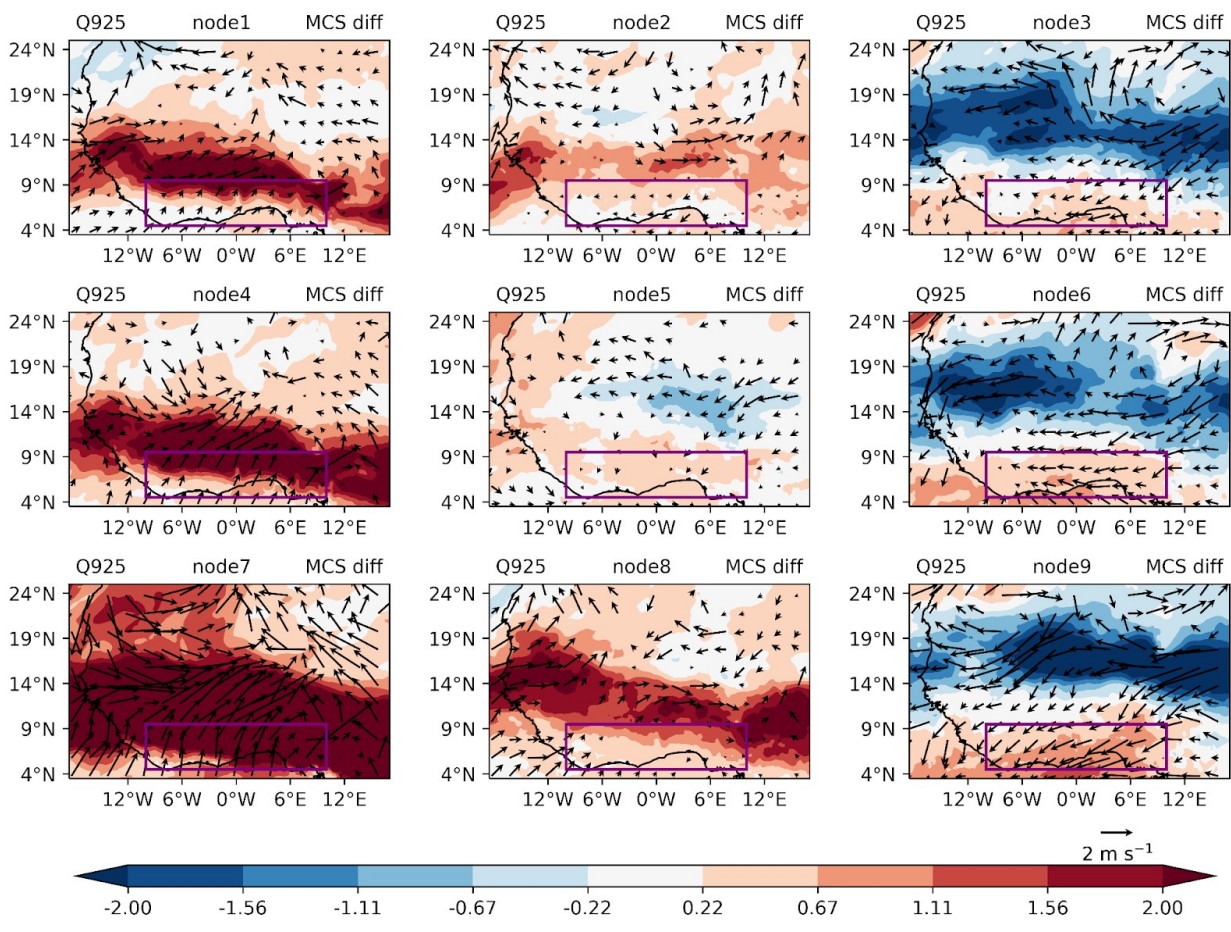



**Figure 9.** 12 UTC MCS-day composite anomalies of specific humidity (shading; g kg$^{-1}$ ) and 925-hPa winds (vectors; m s$^{-1}$ ) in 9 nodes based on SOM analysis. Specific humidity anomalies are shown when they are significant at the 5% level; wind vectors are shown when either the zonal or meridional wind anomalies are significant at the 5% level.

During the dry season nodes (1,4,7), a positive widespread moisture anomaly maximum is observed with anomalous south-westerly winds over SWA (Fig. 9). This depicts a substantial enhancement in the low-level moisture transport as a result of the few days with convective activities during the dry season. In the transition nodes (2,5,8), low-level moisture anomalies during convective activity days show weak and mostly insignificant behaviour along the SWA coast based on the two-sided Student's t-test. In node 8, a positive moisture anomaly is located over the northern part of SWA. During monsoon nodes (3,6,9), a notable region of anomalous low-level easterly wind is observed over the Sahel, indicating a weakening of the south-westerly monsoon winds and of the low-level westerly jet, which reduces moisture transport towards the Sahel. This is evident in the negative moisture anomalies over the Sahel and the increase in moisture over the coastal regions during MCS days, which can result in less convective activities over the Sahel region and more convective activities over coastal areas.

We now consider low-level temperature anomalies to detect potential changes in temperature gradients and SHL strength on MCS days. Figure 10 shows a widespread increase in temperature north of SWA during days with active convection in the dry (1,4,7) and transition (2,8) nodes, which may explain strengthened south-westerly wind anomalies in some of these nodes (c.f. Fig. 9). The SWA region in the dry and transition nodes, on the other hand, reveals a negative and/or insignificant change in temperature during MCS days when compared with the mean climatology. In monsoon nodes 3, 6, and 9, temperatures are enhanced in most parts of West Africa including SWA during days with active convection.

15    15

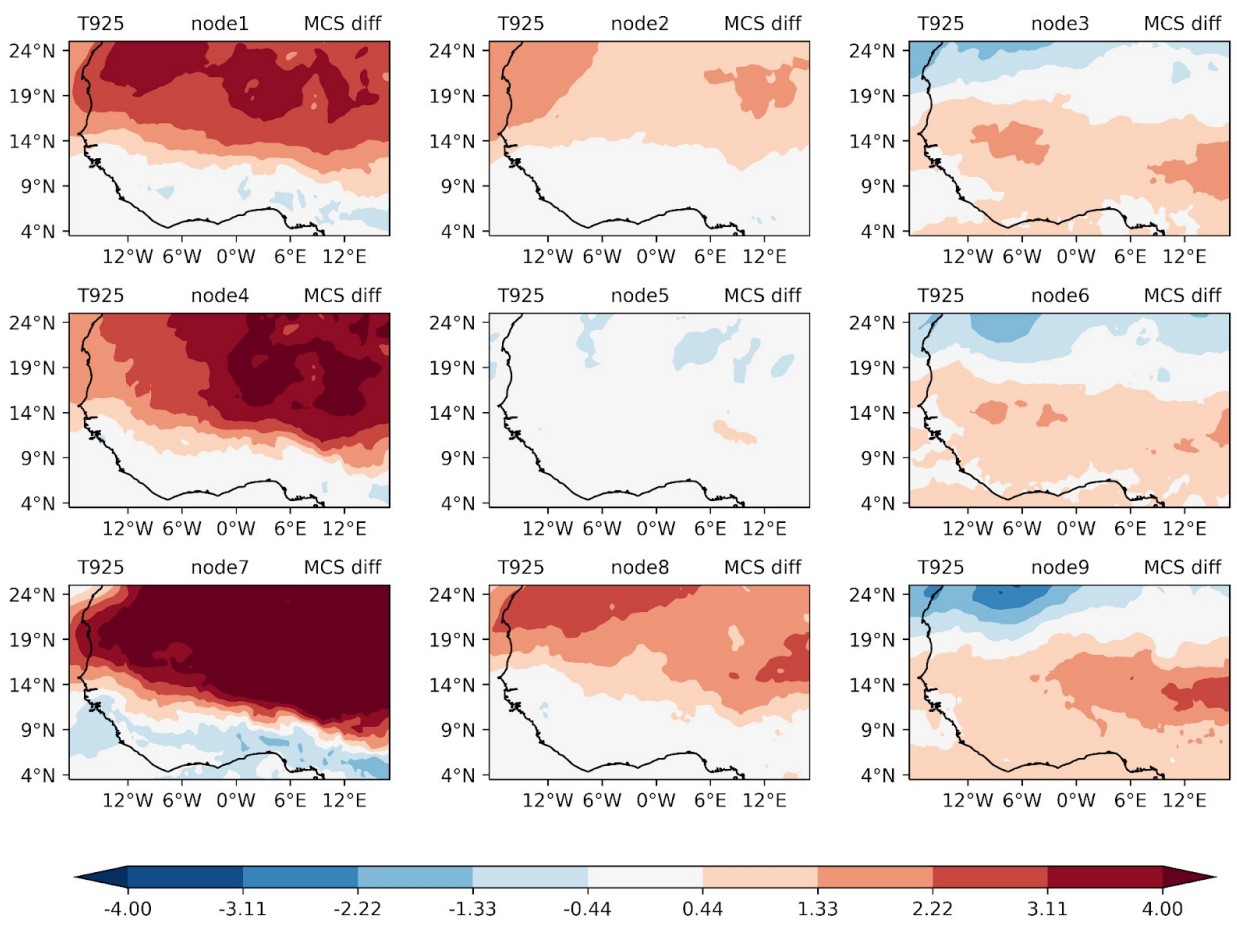

**Figure 10.** 12 UTC composite anomalies of 925hPa temperatures (ºC) in 9 nodes based on SOM analysis. Temperature anomalies are shown when they are significant at the 5% level.

Figure 11 shows the spatial distribution of zonal wind shear anomaly between days with convective MCSs over SWA and the climatological zonal wind shear mean for the 9 different nodes across West Africa. Generally, all dry and transition nodes except node 5, reveal a widespread increase in easterly zonal wind shear anomaly over West Africa with the dry nodes depicting stronger events. Zonal wind shear anomalies tend to be stronger and easterly during the dry season with their peak partly over SWA, but resides to the north of SWA during the transition seasons (nodes 2 and 8). The positive shear anomaly patterns align with patterns of strengthened temperature gradients for respective dry and transition season nodes (c.f. Fig. 10): only node 5 shows no large-scale temperature anomalies and consequently patchy changes in shear, while strongest shear increases occur for node 7 alongside the highest temperature gradient increase. Nodes 2 and 8 experience an appreciably significant increase in easterly zonal wind shear over SWA for MCS days during the transition seasons. The monsoon nodes (3,6,9), on the other hand, exhibit a significant increase in easterly zonal wind shear mainly confined to the south with a pronounced signal in node 9 associated with a peak in eastern-Sahel warming (Fig. 10). In line with the expected zonal wind shear response to an increased large-scale meridional temperature gradient, we thus find the strongest

16    16

easterly zonal wind shear anomalies for nodes with strongest positive low-level temperature anomalies to the north
of SWA (nodes 1,4,7; followed by nodes 2,8), highlighting that a warmer Sahel can promote MCS-favourable shear
conditions in SWA.


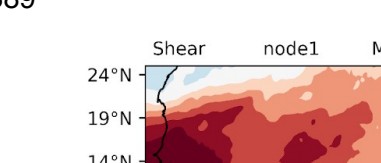
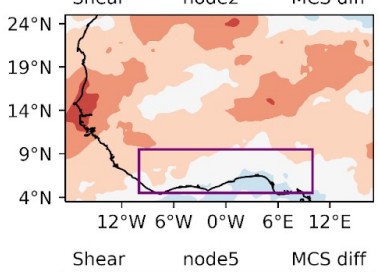
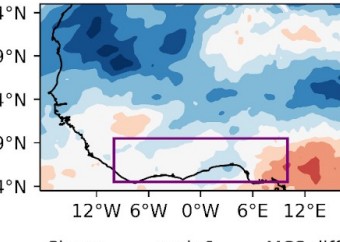
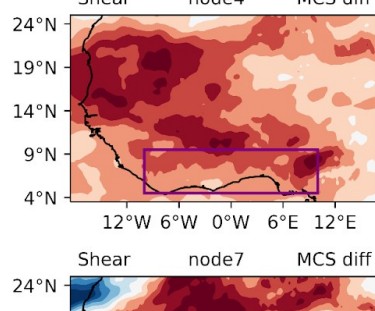
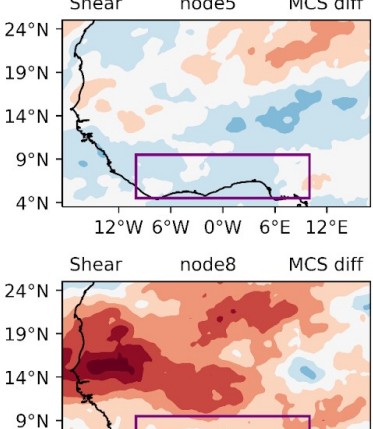
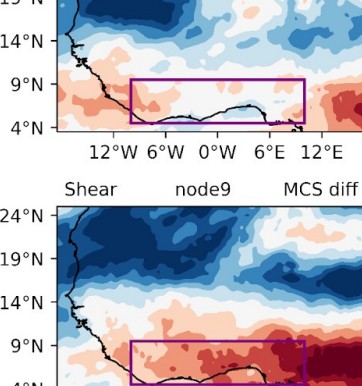
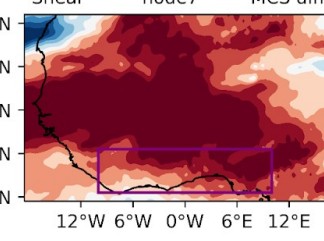
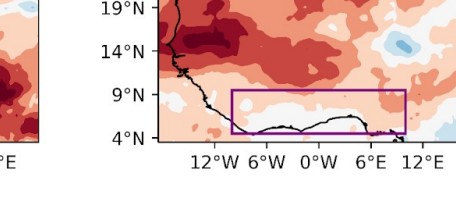


**Figure 11.** 12 UTC composite anomalies of zonal wind shear (m s$^{-1}$) in 9 nodes based on SOM analysis. zonal wind
shear anomalies are shown when they are significant at the 5% level.

Investigating the first-order condition for convection development, we also evaluate CAPE for a parcel at
925 hPa to ascertain the level of increased MCS-day instability in various nodes over SWA (Fig. 12). A large strip
of higher CAPE values extending over the entire region of SWA and the southern Sahel from 5°N−15°N is observed
(dry and transition nodes). This large strip of higher CAPE is situated mainly in central and east of SWA, while part
of the west coast tends to depict patterns of lower CAPE values, suggesting increased MCS likelihood only for the
central and eastern parts of the domain. During monsoon nodes, node 3 shows a broad strip of high CAPE values in
particular to the coast and in some instances extends to the entire SWA (node 6) and north of SWA (node 9). Higher
CAPE conditions over SWA are to differing degrees significantly associated with decreased CAPE in the Sahelian
region, creating a dipole pattern that can occur during transition and monsoon periods according to node frequencies
(cf. Fig 1). Overall, all nodes show positive CAPE and negative convective inhibition (c.f. Supplementary Fig. S4)
anomalies for MCS days in parts of SWA, creating an environment sufficiently unstable to support the development
of convection. The close alignment with regions of increased low-level humidity (Fig. 9) suggests increased low-
level moisture advection as the main driver for these instability changes.

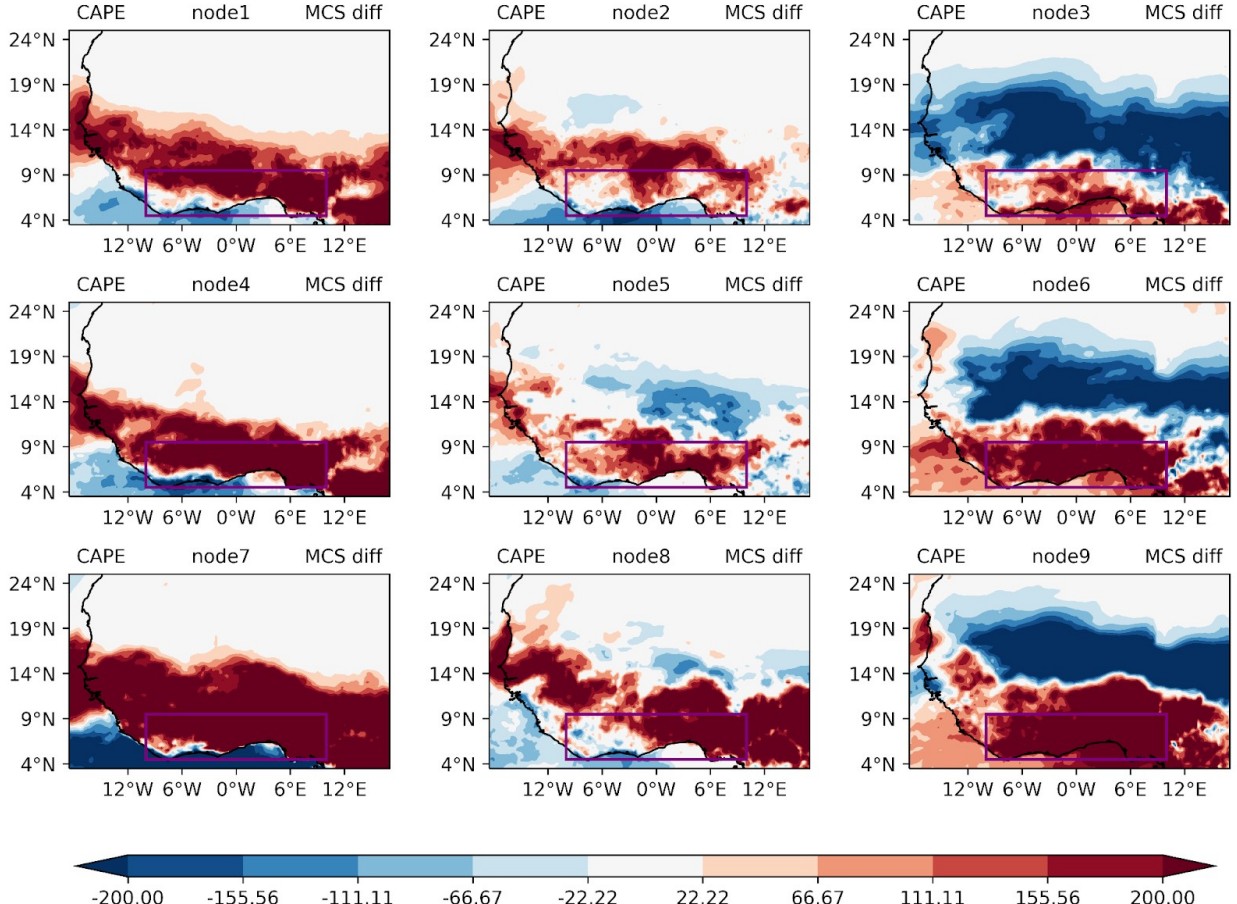


**Figure 12.** 12 UTC composite anomalies of CAPE (J kg$^{-1}$) for MCSs occurring in each type of large-scale
environment determined by the SOM analysis over SWA. CAPE anomalies are shown when they are significant at
the 5% level.


### 4.3 MCS driver variability within nodes

The drivers of MCSs within different nodes are considered to examine their relative importance within the
different large-scale states (Fig. 13), concentrating on total column water vapor (TCWV) and zonal wind shear.
TCWV instead of single-level specific humidity is used here to capture the changes in total moisture available to
MCSs under the different regimes. For this analysis, both atmospheric drivers were sampled locally under pre-
convective conditions at 1200 UTC at the location where MCSs occurred subsequently at 1800 UTC. Dry season
nodes (1,4,7) exhibit the lowest climatological conditions in both wind shear and TCWV. This illustrates the
relatively hostile conditions for storms in the mean for these nodes, predominantly representing dry season
conditions and explaining the low storm frequency of only 0.6-1.7 MCSs per day (cf. Fig. 9). All monsoon nodes
(3,6,9) show on average slightly higher TCWV than transition nodes (2,5,8), but covering a similar range of shear
conditions. Considering MCS day conditions, most nodes feature significantly higher TCWV and shear conditions
relative to the climatological mean node states. Solely for monsoon season nodes (3,6,9), TCWV shows no
significant change, while shear still increases for nodes 6 and 9. Note that while monsoon months feature higher
TCWV and similar shear conditions compared to transition nodes for MCS-location climatologies in Fig. 13, a
larger domain area is affected by MCS-favourable conditions for transition nodes (c.f. Figs. 5,6). As a consequence,
transition nodes exhibit higher overall MCS frequencies. Interestingly, for MCS days, dry season node conditions
even move into the ranges of climatological conditions identified for transition season nodes, though still exhibit the
lowest values in TCWV and zonal wind shear compared to MCS day conditions of transition and monsoon season
nodes.


















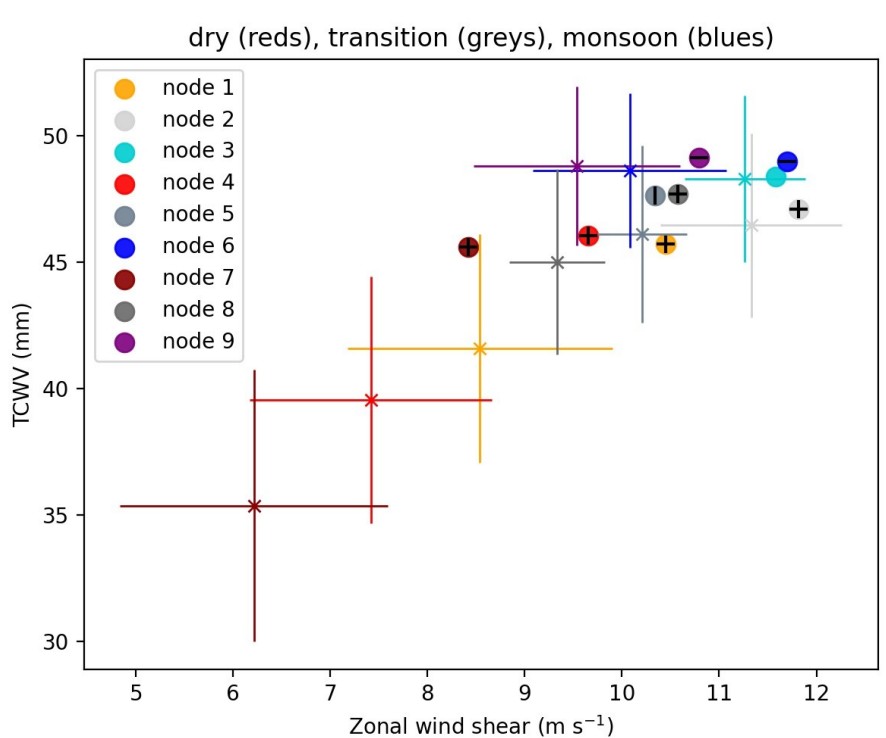

**Figure 13.** Mean node climatologies and MCS-day conditions over SWA. The node climatologies are depicted as
(x) with whiskers extending one standard deviation. Circles denote corresponding mean MCS-day conditions.
Horizontal black lines in the circles indicate significant differences in the shear mean, while a vertical black line
marks a significant difference in the TCWV mean against node climatologies based on Welch's t-tests (p < 0.05)


Generally, it can be noted that all nodes show increased TCWV on MCS days compared to their
climatology. The smallest changes for both TCWV and zonal wind shear between climatology and MCS day occur
for node 3, which has its highest frequency for pre-monsoon transition month May but is still common throughout
the monsoon season (c.f. Fig. 1). Together with node 5, it is also the only node for which zonal wind shear
conditions remain approximately similar, but with climatological zonal wind shear strengths already reaching > 10
m/s at MCS location. Overall, mean node environmental conditions become more similar for MCS-days relative to
the climatologies, illustrating that favourable MCS conditions converge towards high TCWV and high zonal wind
shear environments irrespective of the large-scale situation.
**5 Conclusion**

In this study, we identified nine synoptic states over West Africa and examined what changes are
associated with favourable MCS environments in Southern West Africa under these states. For the definition of
synoptic states and MCS days, we used self-organizing maps (SOM) based on ERA5 925 hPa geopotential height
data and 12 years of MCS imagery using Meteosat Second Generation (MSG) 10.8 $\mu$m-band brightness temperature
data (2004-15), respectively. To investigate how the distinct synoptic states change to support MCS development in
SWA, we compared mean climatological node states to node sub-samples of MCS days in SWA.
We found the identified synoptic states, based on a 3x3 SOM matrix, to exhibit frequency distributions that are
linked to different phases of the West African seasonal rainfall cycle, which we classified as dry (nodes 1, 4, 7),
transition (nodes 2, 5, 8) and monsoon (nodes 3, 6, 9) season, albeit most nodes are not strictly confined to one
season. We found that different nodes identified within one season exhibit key differences in persistence
(consecutive node days) and node succession. Specifically, each season (dry, transition, monsoon) contains a node
that is frequently preceded or followed by a node of another season (nodes 1, 2, 3), as well as a node that
predominantly shows within-season succession (nodes 7, 8, 9). The shortest node persistence of 1.7-1.9 days was
found for nodes 4, 5, and 6. These nodes at the same time represent intermediate synoptic states that develop from or
into a different node of the same season. The SOM methodology thus seems a promising approach to identify states
of variability beyond the established West African monsoon phases (e.g. Thorncroft et al 2011).
In spite of these clear differences in node persistence and succession, large-scale differences in node
climatologies of atmospheric MCS drivers (low-level wind field, 925hPa humidity, and temperature, CAPE) are
most pronounced between nodes of different seasons, while same-season nodes show strong pattern similarities.
Notably, however, MCS-day node anomalies, as compared to full node climatologies, all show clear increases in
low-level humidity and/or wind shear over the SWA region, which are important ingredients for MCS development
(Klein et al. 2021). For dry season nodes, these changes are associated with higher temperatures in the Sahel and
Sahara, driving stronger south-westerly humid winds inland while increasing shear due to an enhanced meridional
temperature gradient on land. Monsoon season nodes on the other hand show the opposite, where a weakening of the
south-westerlies and of the Sahelian low-level westerly jet indicates a southward shift of the monsoon circulation.
This results in more moisture, and for nodes 6, 9 also in higher shear, over SWA, where the latter is linked to a
warmer and drier Sahel during monsoonal southward shifts, creating a dipole pattern. Generally, we find the
strongest MCS-day zonal wind shear anomalies over SWA for nodes with the strongest low-level temperature
anomalies to the north of SWA, representative of favorable MCS conditions in SWA during periods of a warmer

20    20

Sahel. Strengthened wind shear due to a warmer Sahara was previously also identified to drive MCS intensification
in the Sahel (Taylor et al. 2017).
Thus, meridional displacements of the extent to which south-westerly winds from the Atlantic penetrate
inland and the associated positioning of the meridional temperature gradient seems to be key mechanisms by which
MCS days in SWA are created for both, dry and monsoon season node synoptic states. Such meridional
displacements have previously been identified as important drivers of monsoon variability on inter-annual (e.g.
Nicholson and Webster 2008) and intra-seasonal (e.g. Janicot et al. 2011, Talib et al. 2022) timescales. Here, we are
looking at higher-frequency changes with average node persistence between 1.7-4.3 days. Transition nodes show
weaker signals and a mixture of a southward (node 5) or northward (node 8) displaced circulation, which may be
linked to the fact that these nodes predominantly occur in months when the monsoon circulation and its rainfall band
are positioned over SWA (Maranan et al. 2018). Indeed, we find MCSs to be most likely to develop under transition
season node conditions (2.8 MCS/day across SWA domain). There is strong potential for further exploration of the
synoptic differences between transition season nodes and their meridional shifts on MCS days, as these may in some
cases be representative of monsoon onset conditions or a delayed monsoon retreat.
Pre-convective atmospheric anomalies at locations where afternoon development of MCSs took place were
found to be weakest for transition season node 5, lacking significant changes in wind shear, and for monsoon season
nodes 3, 6, 9, for which none showed significant changes in total column moisture, albeit increased moisture at low-
levels contributes to elevated CAPE. Here it should be noted that weak anomalies signify nodes whose mean
climatological conditions already tend to be more favorable for MCS development with respect to that variable, such
that MCS days differ little from the node mean, which, perhaps expectedly, is the case for certain transition and
monsoon rather than dry season nodes.
Generally, however, we find node environmental conditions to become more similar for MCS days relative
to their node climatologies, illustrating that favorable MCS conditions converge towards high TCWV/high zonal
wind shear states. Overall, our results show that MCSs develop on average in high moisture, high zonal wind shear
local environments under all large-scale situations throughout the year. The large-scale situation however defines the
frequency at which favorable MCS environments can occur.

*Code and data availability*. Codes for the findings of this study are available upon reasonable request from the
authors. The processing of ERA5 data made direct access to the primary data archive held at ECMWF, and is
available from the Copernicus Data Store (https://cds.climate.copernicus.eu/) and the MSG data are available from
http://www.eumetsat.int.

*Author contributions*. FN, NABK and CK conceptualized the study, with input from KAQ; All authors contributed
to and discussed the methodological design, and analyses were conducted by FN and CK; FN, ROB and KAQ wrote
the manuscript draft; CK, NABK, PE, GMLDQ and HAK reviewed and edited the manuscript.

*Competing interests*. The contact author has declared that none of the authors has any competing interests.

*Acknowledgments.* This work is supported by a grant from the Government of Canada, provided through Global

Affairs Canada, www.international.gc.ca (accessed on 1 January 2021), and the International Development Research

Centre, www.idrc.ca, (accessed on 1 June 2022) to the African Institute for Mathematical Sciences—Next Einstein

Initiative (AIMS-NEI) [Number: 108246-001]. CK acknowledges funding from the NERC-funded LMCS project

(NE/W001888/1). KQ also acknowledges funding from the National Research Foundation (NRF), South Africa.

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
