# Peer review of "Classification Of Large-Scale Environments That Drive The Formation Of Mesoscale Convective Systems Over Southern West Africa"

_Weather and Climate Dynamics, 2022_

## Referee Comment (RC1)

Review for "Classification of Large-Scale Environments that drive the formation of Mesoscale Convective Systems over Southern West Africa" by Nkrumah et al.

**Overview**

The present manuscript links large-scale atmospheric patterns over West Africa with the occurrence of mesoscale convective systems (MCSs) over the largely understudied southern West African (SWA) region. By applying the self-organizing maps (SOM) technique on a long-term (40-year) ERA5 reanalysis dataset, the authors identified six major atmospheric patterns ("nodes") which broadly represent the rainy and dry seasons of SWA (pre-, peak, post-monsoon). Making use of satellite-based METEOSAT infrared images over 12 years, anomaly patterns from the mean state of the nodes are evaluated to infer typical environmental conditions for the development of MCSs. Here, the authors identified a high-moisture/high-shear environment to be favourable for MCS development under all node situations. While high moisture load is driven by enhanced low-level south-westerlies, stronger shear can be attributed to a warmer Sahel and thus enhanced mid-level easterlies as a consequence of a stronger temperature gradient over West Africa.

As the authors pointed out, drivers of rainfall over the densely populated SWA are still understudied, largely due to the complex and variable interplay between the West African monsoon circulation and local to regional characteristics such as topography or coastal effects. Thus, studies dedicated to SWA like the present manuscript can be a welcome addition and of certain relevance for the scientific community. Overall, I like the paper for its comprehensible presentation and conciseness, however, the latter of which I also see as a shortcoming at places. More details are given in the comments below. Although I do think that, on their own, the comments are largely minor, it can be major if accumulated. Otherwise, I believe the methods used in this study are generally sound. The topic of the manuscript fits within the scope of the journal.

**General comments/questions**

- The authors used 1200 UTC and 1800 UTC, respectively, as the reference times for pre-MCS conditions and the identification of an "MCS day". While this is intuitive in many ways (e.g. highest occurrence frequency of convection in the afternoon/early evening), an entity that is excluded with this definition are nighttime, potentially fast-moving convective systems (e.g. squall lines), which also occur over SWA (Fink and Reiner (2003)) and which can also be important rain contributors in the region (Maranan et al. (2019)). Although I acknowledge that a full analysis exceeds the scope of this study, I believe it is worth including and discussing them (e.g. at 2100 UTC vs 0300 UTC, provided the sample size is high enough), for instance, by extending Fig. 10 by this subset of nighttime convection. I can imagine that, lacking CAPE during the night, these convective systems are increasingly shear-driven.

- Likewise, I do think it is worthwhile to investigate the environmental conditions of "no-MCS" cases. As far as I've understood, the mean-state composites contain all daily timesteps in the 1980-2020 period. Therefore, the anomalies for this case can be integrated in Fig. 10 as well.

- When CAPE is discussed, CIN should be evaluated alongside. This might even be of some relevance for the "no-MCS" cases in the previous bullet point.

- Can the authors explain why they changed from 925 hPa specific humidity in Fig. 3 to TCWV in Fig. 10? Although TCWV is primarily influenced by the evolution of the humidity field in the lower troposphere, I would stay consistent here by choosing one.

- A couple of question regarding MCS data and definitions which may be clarified in the manuscript as well:
    1. Can the authors elaborate why METEOSAT data is limited to 2015?
    2. Have the authors used 15-minute METEOSAT data?
    3. What do the authors mean by "5 MCS snapshots"? The detection of at least 5 MCSs or the detection of MCSs at five timesteps between 1600 and 1900 UTC? If the latter, is an MCS day identified irrespective of the overall number of MCSs at 1800 UTC? So the detection of a single MCS is sufficient?

4. Are only land-based MCSs accepted? What would be the criterion for the position? Center of mass?
5. How did the authors determine the rainfall amount? What dataset is used for this?

- Up until Fig. 6, the extent of the "SWA region" never became clear. The authors may consider including an introductory map of West Africa (e.g. orography) with the SWA region outlined.

- Can the authors clarify in more detail how node 4 and 5 have to be distinguished in the context of the evolution of the WAM? From Fig. 2, the only major difference I can spot is that the background geopotential in node 5 is higher than in node 4. Is that also what the SOM technique identified as the decisive difference to define a dedicated node?

- What atmospheric patterns are shown in the remaining three nodes which were dismissed for this study? How many days were then excluded from the overall sample?

**Specific comments/questions**

L42:      What reference is "Change 2014"?
L76:      "In previous studies that evaluated MCS-favouring atmospheric environments, less attention was given to the importance of large-scale WAM modes and their effect on regional MCS frequencies in SWA". Nonetheless, there are studies that address the large-scale settings for WAM-related rainfall throughout the seasons, e.g. the studies by Sultan and Janicot (see reference). Although they do not refer specifically to MCSs, MCSs are part of the WAM rainfall patterns.
L95:      "For this purpose, a classification using a self organizing map (SOM; Kohonen 2001) analysis was carried out to characterize large-scale WAM patterns during the 1981-2019 period". Any reason why the mapping was performed until 2019, but the analysis of atmospheric fields until 2020?
L109:     Better "137 vertical model levels".
L111:     Can the authors explain what they used the 250 hPa horizontal wind for? Have the authors also investigated the Tropical Easterly Jet? In any case, the 250 hPa wind was never addressed anymore.
L129-133: Might be better to shift this to the introduction.
L168:     Seasonal cycle of what? Monthly rainfall amount?
L185:     Do you mean "low pressure"?
L214:     "…show significant changes over the last 4 decades". In what way exactly?
Sec. 4.2: Have the anomalies been calculated from the mean state in Fig. 2, i.e. based on 1981-2020? Since the MCS days run from 2004-2015, have the authors account for potential trends between the periods 1981-2003 and 2004-2015?
Figure 5:  A bit surprising to find zero MCSs in February, but probably a consequence of the high areal MCS criterion chosen in this study.
L227:     Again, the definition of the SWA domain needs to be outlined earlier.
Figure 6:  Again, does "location of the MCS" refer to the center of mass of the cloud area? Does the MCS frequency refer to the amount of MCS days compared to the total number of node days? Does that explain why there are much more MCS dots for node 6 than node 5, the latter of which has a higher frequency?
L248:     What do the authors mean by "insignificant behaviour"?
L250:     Also seemingly partly northerlies from the Mediterranean region.
Figure 8:  Can the authors add the maps for the mean-state of vertical wind shear in section 4.1 and discuss them for more clarity?
Figure 9:  As mentioned, CIN should be shown and discussed as well.
L309:     "…illustrating the relatively storm-hostile mean conditions…". But doesn't the mean state include all time steps, including MCS days? As outlined in the general comments, the authors may add the specific non-MCS state in Fig. 10 for clarity.
Figure 10: The reddish colours are hard to distinguish.

**References**

Fink, A. H., & Reiner, A. (2003). Spatiotemporal variability of the relation between African easterly waves and West African squall lines in 1998 and 1999. Journal of Geophysical Research: Atmospheres, 108(D11).

Maranan, M., Fink, A. H., Knippertz, P., Francis, S. D., Akpo, A. B., Jegede, G., & Yorke, C. (2019). Interactions between convection and a moist vortex associated with an extreme rainfall event over southern West Africa. Monthly Weather Review, 147(7), 2309-2328.

Sultan, B., Janicot, S., & Diedhiou, A. (2003). The West African monsoon dynamics. Part I: Documentation of intraseasonal variability. Journal of Climate, 16(21), 3389-3406.

Sultan, B., & Janicot, S. (2003). The West African monsoon dynamics. Part II: The "preonset" and "onset" of the summer monsoon. Journal of climate, 16(21), 3407-3427.

---

## Referee Comment (RC2)

The manuscript analyses large-scale synoptic conditions in West Africa, and aims to identify flow anomalies related to the occurrence of MCS in a smaller domain of South West Africa (SWA). The study relies on daily ECMWF ERA5 data for the period of 1981-2020 for the large-scale environment and cloud-top temperatures retrieved from satellite data to identify MCS events. The 925 hPa geopotential surface is classified using SOM, and 6 nodes out of 9 are discussed. The nodes correspond to the seasonal evolution of the West-African Monsoon (WAM). The flow features are discussed with an emphasis on winds, humidity, vertical zonal-wind shear, and CAPE. Nodal variations corresponding to MCS events are analyzed regionally and locally and solidify the link between increased TCWV and vertical zonal-wind shear to MCS-favoring conditions in the SWA. It concludes that local increases in wind shear and humidity are a common MCS feature for all MCS days, with strong wind shear correlated to large CAPE and dictating the onset of MCS. However, since this mechanism has been consolidated for the Sahel, there seems to be room for a deeper investigation here.

MCS studies dedicated to the SWA region are lacking, and the motivation for the present one is clear. The introduction is well structured and written for the most part, though some elaboration is required when it comes to the SOM classification. The methodological approach appears novel and highly practical. The results are clear and concise, perhaps to a fault – some aspects could be elaborated. Still, the results enable the conclusions drawn by the authors.

The subject of the manuscript is relevant and fits the journal's scope. However, there are major issues that should be addressed before consideration of this work for publication. I summarize the main issues in the following, along with more specific comments and suggestions for addressing the main concerns and improving clarity.

**General comments**

1) The SOM analysis should be presented as a whole, and the reason for rejecting certain nodes from the analysis should be better explained. No discussion is dedicated to the SOM configuration, primarily the choice of the number of nodes, but also other SOM parameters (neighborhood size, topology, initial coverage space, etc.). The robustness of the SOM clustering is not evaluated. A significance test should be added for the detected geopotential patterns, and the SOM errors (quantification and topological) should be discussed. Furthermore, the low number of nodes under consideration for this study appears to not fully justify a SOM analysis in the first place. Seeing as each node roughly corresponds to a certain season and is treated as a seasonal mean, it appears that the information presented here can be yielded by a simple seasonal decomposition. Ideas to enrich the SOM analysis and the gain from it can be found in the literature quoted by the authors. Otherwise, the authors may consider replacing the SOM analysis with a simple seasonal decomposition. Please also refer to more specific comments in this regard, below,

and the following highly relevant references with very similar motivations and methodologies:

- Liu, Y., Weisberg, R. H., and J. I. Mwasiagi (Eds.): A review of self-organizing map applications in meteorology and oceanography, Self-Organizing Maps: Applications and Novel Algorithm Design, InTech publications, Rijeka, Croatia, 2011.

- Gueye AK, Janicot S, Niang A, Sawadogo S, Sultan B, Diongue-Niang A, Thiria S 2010 Weather regimes over Senegal during the summer monsoon season using self-organizing maps and hierarchical ascendant classification. Part I: synoptic time scale. Climate dynamics. doi:10.1007/s00382-010-0782-6

- Espinoza, J. C., Lengaigne, M., Ronchail, J., and Janicot, S.: Largescale circulation patterns and related rainfall in the Amazon Basin: a neuronal networks approach, Clim. Dynam., 38, 121–140, https://doi.org/10.1007/s00382-011-1010-8, 2012

- Givon, Y., Keller Jr, D., Silverman, V., Pennel, R., Drobinski, P., & Raveh-Rubin, S. (2021). Large-scale drivers of the mistral wind: link to Rossby wave life cycles and seasonal variability. Weather and Climate Dynamics, 2(3), 609-630.

2) The choice of low-level geopotential heights as a clustering agent should be better motivated, given the relatively low correspondence between it and the low-level winds in the domain, which are described as the main process driver throughout the manuscript. Have the authors considered directly classifying the wind field?

3) Nodal trends – this section appears unrelated to the motivations of the paper and is very slim. I suggest a deeper analysis to explore, for instance, corresponding trends in MCS events. Otherwise, consider removing this section.

4) MCS data – I think this data should be further explored. For one, it can be better presented using a density plot. Secondly, spatial variability should be discussed and possibly explained, with an emphasis on variations between nodes and seasons within the nodes. Finally, it's worth checking for MCS behavior on off-season node days.

5) The link to predictability can be improved. For instance, can we learn anything from a lagged correlation between nodal transitions and MCS density?

**Specific comments:**

L24: Too vague. What is the input used for classification? i.e., how do you define a "synoptic circulation-type"?

L32: Unclear. Do you mean vertical/ horizontal wind shear? what is the field under discussion here?

L35: The use of the term "shear" or "wind shear" when alternatively referring to vertical and zonal shear is confusing. You should specify which shear is under consideration throughout the paper.

L49: Missing a link to WAM. The change of subject is too sudden and does not flow from the previous paragraph. Consider opening the section with lines 53-54

L94: "large-scale patterns" - Too vague. You should name the parameter used for the classification here.

L121: "SWA domain" - This domain should either be specified in latitude and longitude boundaries or displayed in a figure earlier on. Possibly both.

L124: This section requires more detail.
For instance, what is the SOM topology? It would be useful to add a neighbor distances map and to evaluate SOM errors. The number of members in each cluster should also be given, preferably in Fig 1.

L128-129: This statement is true for many optional classification inputs. In the present study, the focus is on the tropics where geostrophic balance is not obvious, as seen by your results. Therefore, the choice to classify patterns using geopotential heights should be justified.

L131: Each method has its advantages and disadvantages, and each can be more suited for a different study. Refrain from making conclusive statements.

L132: "data is not discretized and orthogonality is not forced" – Again, these are not clear advantages. The SOM's strengths and weaknesses should be discussed in the context of the present study.

L142: More information is required on what led to the choice of 9 clusters. Have you evaluated the network errors under the different configurations (SOM size and other parameters) to show that 9 is the most compatible?

L153: Why not compare to non-MCS days within the node? This may highlight the signal you are after.

L154: regarding the T-test – on Which confidence level was it conducted? have you used any method to detect false positives in the multi-gridded test? See Wilks 2016 for example.

Wilks, D.: "The stippling shows statistically significant grid points": How research results are routinely overstated and over-interpreted, and what to do about it, B. Am. Meteorol. Soc., 97, 2263–2273, https://doi.org/10.1175/BAMS-D-15-00267.1, 2016.

L165: Why is the complete SOM not shown? This is not clear.
If you choose to discard nodes altogether, you should show the full SOM map (9 nodes) first, then explain why not all nodes are relevant, and which ones were removed. The resulting 6-node SOM map should be shown in the context of the full SOM map, as the node locations on the SOM map are crucial for the SOM interpretation. This also raises the question: are the panels in Figures 2-9 arranged correctly? i.e., are neighboring nodes in these Figures also neighbors in the full SOM map?
I suggest repeating the analysis for 6 nodes if that's what you end up analyzing, while completely removing irrelevant dates from the SOM input.

L167-168: Even if some nodes are ignored, the numbering of the nodes should be as in the full SOM analysis, to be consistent with the complete SOM map.

Figure 1: Add the total number of members in each node. Consider normalizing per year and not per month.

Figure 2: Grey grid can be removed to improve visibility. Also, be consistent with X-label intervals. Clarify whether these are daily means or 12 UTC composite.

Figure 3: The low correspondence between winds and geopotential heights in the tropical region raises the question: what is the value of classifying by geopotential if it's not indicative of the flow field? Why not directly classify the velocity/ wind-speed fields?

L211-218: This subsection is too slim. Either remove it or expand it to get to a conclusion .
At the present state, this subsection does not contribute to the main motivation of this study and possibly draws the reader's attention from the main storyline.

Figure 6: This domain should be shown earlier when first presenting the SWA domain.

L258: unclear. Why does high humidity lead to cooling?

L279: This point was given as a well-known fact in the introduction, so I don't see what is the novelty here.

L290: "eastern patterns" – This is not evident in figure 6. This issue should be discussed

L140: "pure node analysis" – What do you mean by this?

**Technical corrections**
L25: which=that.

L38: variabilities=variability.

L42: "Change, 2014" is not a reference, or is missing from the reference list.

L88: environments= parameters?

L97: Is this a correct use of the word stratify? Seems confusing to me. How about grouped/ separated?

L108: product= data source.

L126: daily=daily mean.

L182: SOMs=SOM.

L200: "much more strengthened" – Rephrase. Consider "Intensified", "Increased" and so on.

L211: "A further" = Further.

L213: during=within.

Figure 4: The term "moving mean" seems more fitting.

L225: This second subtitle is redundant.

L244: Repetitive.

Figure 7: The colors appear saturated. Expand the color map beyond 2K to avoid this.

L274: observes= demonstrates/ exhibits.

L290: observe = show/ depict.

L308-311: Long sentence, consider splitting.

L323: "making node 2…" This sentence is unclear, rephrase.

L330: "This season" – Unclear which season is that.

---

## Author Comment (AC1)

**RESPONSE TO RC1**

**Referee #1**: We are grateful for the in-depth comments and specific analysis advice the referee provided for our manuscript. This helped significantly improve our results' robustness and sharpen our discussion. Below, we give a point-by-point response to the comments with line references to the new manuscript version. We also provide a tracked changes document highlighting the differences from the previous version.

**General comments/questions:**
- The authors used 1200 UTC and 1800 UTC, respectively, as the reference times for pre-MCS conditions and the identification of an "MCS day". While this is intuitive in many ways (e.g. highest occurrence frequency of convection in the afternoon/early evening), an entity that is excluded with this definition is nighttime, potentially fast-moving convective systems (e.g. squall lines), which also occur over SWA (Fink and Reiner (2003)) and which can also be important rain contributors in the region (Maranan et al. (2019)). Although I acknowledge that a full analysis exceeds the scope of this study, I believe it is worth including and discussing them (e.g. at 2100 UTC vs 0300 UTC, provided the sample size is high enough), for instance, by extending Fig. 10 by this subset of nighttime convection. I can imagine that lacking CAPE during the night, these convective systems are increasingly shear-driven.

Thank you for the suggestion. We agree that nighttime convective systems are important contributors to rainfall over the entire West African region. Nonetheless, this work focused on the time when the frequency of MCSs reaches a maximum (e.g. Mathon and Laurent 2001).

- Likewise, I do think it is worthwhile to investigate the environmental conditions of "no-MCS" cases. As far as I've understood, the mean-state composites contain all daily timesteps in the 1980-2020 period. Therefore, the anomalies for this case can be integrated in Fig. 10 as well.

The scope of the work looked at investigating environmental conditions favourable for MCSs over the region. Considering "no-MCS" events is out of the scope of this work but can be looked at to improve this work as new research.

- When CAPE is discussed, CIN should be evaluated alongside. This might even be of some relevance for the "no-MCS" cases in the previous bullet point.

Thank you for the suggestion. We agree with the referee on this. Here, we only focussed on variables that contribute to instability in the atmosphere and also lead to a stronger updraft. That is why we did not consider CIN since CIN inhibits the formation of deep convection.

- Can the authors explain why they changed from 925 hPa specific humidity in Fig. 3 to TCWV in Fig. 10? Although TCWV is primarily influenced by the evolution of the humidity field in the lower troposphere, I would stay consistent here by choosing one.

As pointed out by the referee, we considered the TCWV due to its ability to represent the total gaseous water in the vertical column of the atmosphere which is influenced by the evolution of the humidity field. TCWV represents the precipitable water the atmosphere holds better than the humidity. We, therefore, had to show both since in the first instance we were looking at an environment that is suitable for instabilities in the atmosphere, of which humidity forms a part.

- A couple of questions regarding MCS data and definitions which may be clarified in the manuscript as well:
  - 1. Can the authors elaborate why METEOSAT data is limited to 2015?

Actually, the METEOSAT data is up to 2018. The limit was chosen to cover the period over which we could identify match-ups between microwave rainfall estimates from IMERG dataset and MCSs between 1800 and 2100 UTC. The IMERG data only starts in the 2000s, and since we wanted MCSs that produce precipitation of a minimum of 5mm, we had to consider MCS with overlapping maximum rainfall pixels. The period 2004 - 2018, therefore, gives us a good representation of rain-producing MCSs.

  - 2. Have the authors used 15-minute METEOSAT data?

Yes. The authors used 15-minute METEOSAT data, and this has been added to the manuscript for clarity.

The Meteosat Second Generation (MSG) cloud-top temperature data, which are available every 15 minutes from the Eumetsat archives online (https://navigator.eumetsat.int/product/EO:EUM:DAT:MSG:HRSEVIRI) was used in this study.

  - 3. What do the authors mean by "5 MCS snapshots"? The detection of at least 5 MCSs or the detection of MCSs at five timesteps between 1600 and 1900 UTC? If the latter, is an MCS day identified irrespective of the overall number of MCSs at 1800 UTC? So the detection of a single MCS is sufficient?

Thank you for the opportunity to clarify. The "5 MCS snapshots" which have now been changed to "5 MCSs" is the detection of at least 5 MCSs between 1600 and 1900 UTC per day and not 5 timesteps. This means we expect to detect at least 5 MCSs within the timeframe to call it an MCS day.

  - 4. Are only land-based MCSs accepted? What would be the criterion for the position? Center of mass?

We considered only the land-based MCSs because MCSs over land are fundamentally more intense and deep than its counterpart over the ocean (Mohr and Zipser 1996). This has been included in the manuscript to give a clear explanation.

Here, only land-based MCSs because MCSs over land are fundamentally more intense and deep than its counterpart over the ocean (Mohr and Zipser 1996).

  - 5. How did the authors determine the rainfall amount? What dataset is used for this?

The rainfall amount was determined from rainfall snapshots of the "high-quality precipitation" (HQ) a field within the Integrated Multi-satellite Retrievals for Global Precipitation Measurement (IMERG; Huffman et al. 2019) dataset. This has been included in the manuscript as follows:

This can include the same MCS at several timesteps in a day. Corresponding rainfall snapshots were sampled from the "high-quality precipitation" (HQ) field within the Integrated Multi-satellite Retrievals for Global Precipitation Measurement (IMERG; Huffman et al. 2019) dataset.

- Up until Fig. 6, the extent of the "SWA region" never became clear. The authors may consider including an introductory map of West Africa (e.g. orography) with the SWA region outlined.

Thank you for the notification. The "SWA domain" is now shown earlier in Fig. 2 with coordinates clearly shown in the caption of Fig. 2.

- Can the authors clarify in more detail how nodes 4 and 5 have to be distinguished in the context of the evolution of the WAM? From Fig. 2, the only major difference I can spot is that the background geopotential in node 5 is higher than in node 4. Is that also what the SOM technique identified as the decisive difference to define a dedicated node?

Find the answer to the comment above. Comparing nodes 4 and 5 in Fig. 1, it can be clearly seen that the frequency of node cases in Fig. 1 largely occurs during the months of September, October, and November. It can also be seen in node 4 that the frequency in node cases persists in the pre-monsoon season as well as the post-monsoon, with a similar number of cases for some months in both pre-and post-monsoon seasons, although more prominent in the post-monsoon season.

Yes! From Fig. 2, that is what the SOM technique identified as the decisive difference between nodes 4 and 5. That clearly means the background geopotential during the post-monsoon season itself is high as in Fig. 5. The presence of persisting node cases observed in the pre-monsoon season in node 4 reduces the effect of the background geopotential for events in that node.

- What atmospheric patterns are shown in the remaining three nodes which were dismissed for this study? How many days were then excluded from the overall sample?

Thank you for the opportunity to clarify. In choosing the size of the SOM, we considered the distinctiveness and robustness of the circulation systems. A too-small node size reduces the robustness of capturing the predominant spatial characteristics and a too-large node size introduces redundant nodes. The 9-node SOM was chosen as a compromise on states not being overly generalized while capturing the dominant spatial characteristics over the region. However, three (3) nodes were observed to be similar but low in frequency when compared to the other nodes. These 3 similar nodes were combined with the other nodes showing similar features.

In effect, no nodes were dismissed but rather they were grouped with the ones that are obviously quite similar in atmospheric patterns and seasonal frequency.

**Specific comments/questions:**
- L42: What reference is "Change 2014"?

Thank you for pointing this out. We have corrected this reference by replacing it with "IPCC 2014", with its corresponding reference added accordingly as IPCC: Climate Change, 2014: Synthesis Report. Contribution of Working groups I, II, and III to the Fifth Assessment Report of the Intergovernmental Panel on Climate Change [Core Working Team, R.K. Pachauri and L.A. Meyer (eds)]. IPCC, Geneva, Switzerland, 151, 2014.
.
- L76: "In previous studies that evaluated MCS-favouring atmospheric environments, less attention was given to the importance of large-scale WAM modes and their effect on regional MCS frequencies in SWA". Nonetheless, there are studies that address the largescale settings for WAM-related rainfall throughout the seasons, e.g. the studies by Sultan and Janicot (see

reference). Although they do not refer specifically to MCSs, MCSs are part of the WAM rainfall patterns.

Here we absolutely agree and the manuscript highlights that as well that previous studies address the large-scale settings for WAM-related rainfall. The focus of this manuscript was on suitable conditions favourable to the changes in the frequency of MCSs over SWA. Nonetheless, we have rephrased the statement in question to encapsulate the suggestion above.

- L95: "For this purpose, a classification using a self organizing map (SOM; Kohonen 2001) analysis was carried out to characterize large-scale WAM patterns during the 1981-2019 period". Any reason why the mapping was performed until 2019, but the analysis of atmospheric fields until 2020?

Corrected. The mapping was performed until 2020 and the same was used for the analysis.

- L109: Better "137 vertical model levels".

Corrected as pointed out

- L111: Can the authors explain what they used the 250 hPa horizontal wind for? Have the authors also investigated the Tropical Easterly Jet? In any case, the 250 hPa wind was never addressed anymore.

We removed the 250 hPa wind level since it was not used anywhere in the manuscript.

- L129-133: Might be better to shift this to the introduction.

We agree with this suggestion and have therefore moved this statement to the introduction.

- L168: Seasonal cycle of what? Monthly rainfall amount?

It is the seasonal cycle of monthly rainfall amounts as suggested.

- L185: Do you mean "low pressure"?

Here, we were talking about the West African heat low (WAHL) which is known to be a region of high surface temperatures and low surface pressures. It is, therefore, a low-pressure region as stipulated and forms part of the West African monsoon system.

- L214: "…show significant changes over the last 4 decades". In what way exactly?

Thank you for the opportunity to clarify. Based on a mann-kendall trend test conducted along with a test of significance, it is clear that trends observed in Nodes 4 and 5 are significant with p-values of below 0.05 in both nodes.

- Sec. 4.2: Have the anomalies been calculated from the mean state in Fig. 2, i.e. based on 1981-2020? Since the MCS days run from 2004-2015, have the authors account for potential trends between the periods 1981-2003 and 2004-2015?

Thank you. We did not account for potential trends between the two periods. Looking at the period each of them covers (ie. mean state and MCS days), we assume they represent the general behaviour of trends in each state and therefore no need to account for any differences.

- Figure 5: A bit surprising to find zero MCSs in February, but probably a consequence of the high areal MCS criterion chosen in this study.

The focus of MCSs over the study area in this study is during the rainfall season of the SWA domain which mainly starts in March and ends in November. February recorded zero because it wasn't considered in the frame of this work.

- L227: Again, the definition of the SWA domain needs to be outlined earlier.

Done. The SWA domain has been outlined earlier in Fig. 2.

- Figure 6: Again, does "location of the MCS" refer to the center of mass of the cloud area? Does the MCS frequency refer to the amount of MCS days compared to the total number of node days? Does that explain why there are much more MCS dots for node 6 than node 5, the latter of which has a higher frequency?

Here we agree to the view above.

- L248: What do the authors mean by "insignificant behaviour"?

In the manuscript, all anomalies show only regions that are significant at the 5% level. Areas with "insignificant behaviour" are where the two-sided Student's t-test depicts insignificant differences between node climatologies and MCS-day sub-samples.
We have added a phrase to that statement to clarify this point.

- L250: Also seemingly partly northerlies from the Mediterranean region.

The suggestion has been added to the manuscript.

- Figure 8: Can the authors add the maps for the mean-state of vertical wind shear in section 4.1 and discuss them for more clarity?

We have added the maps for the mean-state of zonal wind shear and discussed it accordingly as shown below:

A further investigation was conducted to ascertain the spatial distribution of mean zonal wind shear over SWA (Fig. 4). The patterns demonstrate northward transport during the propagation of the WAM cycle and a wider spread of zonal wind shear as it moves further inland (nodes 1, 2, and 3). These patterns closely follow the southern boundary of weaker geopotential heights representative of high-pressure areas (Fig. 2). During the monsoon season (node 6), zonal wind shear lies clearly to the north of the SWA domain. A southward retreat of zonal wind shear is observed during the post-monsoon season (nodes 4 and 5). Generally, the presence of zonal wind shear can be seen as a necessary condition in the WAM system.

[Figure]

**Figure 4.** 12 UTC composites of zonal wind shear in six nodes based on SOM analysis.

● Figure 9: As mentioned, CIN should be shown and discussed as well.
As said earlier, CIN is not considered in the manuscript because we concentrated on parameters that create instability and promote strong updraft.

● L309: "…illustrating the relatively storm-hostile mean conditions…". But doesn't the mean state include all time steps, including MCS days? As outlined in the general comments, the authors may add the specific non-MCS state in Fig. 10 for clarity.
Thank you for pointing this out. We have corrected the statement.

● Figure 10: The reddish colours are hard to distinguish.
We agree with this assertion and have addressed it accordingly.

---

## Author Comment (AC2)

**RESPONSE TO RC2**

**Referee #2:** We thank the referee for their valuable input, which helped to improve the clarity of our manuscript and figures. The line references below refer to the newly changed manuscript. Additionally, we provide a document with tracked changes.

**General comments**:
1) The SOM analysis should be presented as a whole, and the reason for rejecting certain nodes from the analysis should be better explained. No discussion is dedicated to the SOM configuration, primarily the choice of the number of nodes, but also other SOM parameters (neighborhood size, topology, initial coverage space, etc.). The robustness of the SOM clustering is not evaluated. A significance test should be added for the detected geopotential patterns, and the SOM errors (quantification and topological) should be discussed. Furthermore, the low number of nodes under consideration for this study appears to not fully justify a SOM analysis in the first place. Seeing as each node roughly corresponds to a certain season and is treated as a seasonal mean, it appears that the information presented here can be yielded by a simple seasonal decomposition. Ideas to enrich the SOM analysis and the gain from it can be found in the literature quoted by the authors. Otherwise, the authors may consider replacing the SOM analysis with a simple seasonal decomposition. Please also refer to more specific comments in this regard, below, and the following highly relevant references with very similar motivations and methodologies:
- Liu, Y., Weisberg, R. H., and J. I. Mwasiagi (Eds.): A review of self-organizing map applications in meteorology and oceanography, Self-Organizing Maps: Applications and Novel Algorithm Design, InTech publications, Rijeka, Croatia, 2011.
- Gueye AK, Janicot S, Niang A, Sawadogo S, Sultan B, Diongue-Niang A, Thiria S 2010 Weather regimes over Senegal during the summer monsoon season using self-organizing maps and hierarchical ascendant classification. Part I: synoptic time scale. Climate dynamics. doi:10.1007/s00382-010-0782-6
- Espinoza, J. C., Lengaigne, M., Ronchail, J., and Janicot, S.: Largescale circulation patterns and related rainfall in the Amazon Basin: a neuronal networks approach, Clim. Dynam., 38, 121–140, https://doi.org/10.1007/s00382-011-1010-8, 2012
- Givon, Y., Keller Jr, D., Silverman, V., Pennel, R., Drobinski, P., & Raveh-Rubin, S. (2021). Large-scale drivers of the mistral wind: link to Rossby wave life cycles and seasonal variability. Weather and Climate Dynamics, 2(3), 609-630.

The comments and suggested references are well received. However, we will like to address these comments as follows:

First, we do state that we use a 9-node SOM however, 6-nodes are representative of the West African monsoon pattern and that is why we present the 6-node states. We should mention that to obtain the best SOM, the third stage of the SOM process evaluates the quantization and topological error. An optimal SOM is obtained when the average Euclidean distance is the minimum (the quantization error is the smallest) and when the proportion of all data vectors for which the first and second best matching units are not adjacent is also minimum (the topological error is the lowest). Once the average quantization error has been minimized, the relationships between the predictor and node data are investigated.

Third, we do not agree with using a simple seasonal decomposition method for this analysis as the SOM has clear strengths when compared with a simple seasonal decomposition. For instance, a simple seasonal

decomposition may not identify the combined or mixed pre/post-monsoon states (secondary states) and is likely to identify states as either pre or post-monsoon (primary states), and we will be clearly losing vital information (e.g., Rousi et al. 2015).

In testing for the significance of the identified SOM states, there are several studies (e.g., Hewitson and Crane, 2002; Rousi et al. 2015; Espinoza et al 2012) that support our methodology on the fact that the SOM methodology is data-dependent and such the dominant patterns are representative of the data, thus in the current study a significance test is not necessarily needed. Also, the initial coverage space is mentioned in the data section (domain). We must also highlight here that the SOM is a neural network algorithm as clearly shown in the reviewer's suggested literature (Espinoza et al. 2012).

Nonetheless, we have made additions to the methodology section to provide further clarity and to reflect the reviewer's suggestions.

In this study, the SOM is randomly initialized allowing for hidden patterns and structure in the geopotential height at 925 hPa to be discovered while the algorithm iteratively updates the weights of the nodes to better represent the data. The strength of initializing the SOM this way lies also on its robustness to noise and outliers as a result of the algorithm applying a competitive learning structure to the data which then allows for the formation of distinct clusters. The SOM_PAK algorithm allows the SOM process to minimize quantization and topological errors at the mapping stage when choosing the best SOM as outlined in Lennard and Hegerl (2014). However, there is a trade-off when choosing the size of the SOM, as this is dependent on the need to generalize circulation states for analyses or the need to capture predominant spatial characteristics that affect the local climate. Thus, in this study, we have tested several sizes of the SOM and have arrived at using a 9-node SOM. As depicted in Fig. S1 for a 9-node SOM, it is evident that some nodes are still redundant, and this is a compromise on states not being overly generalized while capturing the dominant spatial characteristics over the region. Here, we agree on six nodes, which allow distinct synoptic states to be reproduced while grouping nodes that are similar. This grouping was done based on similarities in atmospheric patterns and seasonal frequency from the 9-node case.

2) The choice of low-level geopotential heights as a clustering agent should be better motivated, given the relatively low correspondence between it and the low-level winds in the domain, which are described as the main process driver throughout the manuscript. Have the authors considered directly classifying the wind field?
The choice of low-level geopotential height was made in this manuscript because we wanted to consider somehow the influence of the West African Heat Low (WAHL) in influencing instability. At low levels, the geopotential height well describes the strength of the WAHL (Lavaysse et al. 2009; Biasutti et al. 2009).

3) Nodal trends – this section appears unrelated to the motivations of the paper and is very slim. I suggest a deeper analysis to explore, for instance, corresponding trends in MCS events. Otherwise, consider removing this section.
Thank you. This section has been removed.

4) MCS data – I think this data should be further explored. For one, it can be better presented using a density plot. Secondly, spatial variability should be discussed and possibly explained, with an emphasis on variations between nodes and seasons within the nodes. Finally, it's worth checking for MCS behavior on off-season node days.

Thank you for this great suggestion. We consider this suggestion to be an added value to this idea of MCS's impact on the climate of SWA. However, this manuscript tries to understand the conditions surrounding/ favorable to the formation of MCSs. Therefore, we concentrated on the mean position of MCS. Further research work we will consider in the future will pay more attention to the spatial variability of MCSs.

5) The link to predictability can be improved. For instance, can we learn anything from a lagged correlation between nodal transitions and MCS density?

Thank you for the suggestion. We have considered the suggestion and we will be interested to work on that in our future research which will improve the link to predictability.

**Specific comments:**
- L24: Too vague. What is the input used for classification? i.e., how do you define a "synoptic circulation-type"?

The 925hPa geopotential height is used as input to train the SOM. The archetypal modes of the geopotential height obtained are used to describe the characteristic circulations over the region.

- L32: Unclear. Do you mean vertical/ horizontal wind shear? what is the field under discussion here?

Here we talk about the zonal wind shear. We have made changes to the manuscript to reflect the exact field under discussion

- L35: The use of the term "shear" or "wind shear" when alternatively referring to vertical and zonal shear is confusing. You should specify which shear is under consideration throughout the paper.

Throughout the manuscript, wind shear or shear is used to represent zonal wind shear. We have therefore replaced "wind shear and shear" with "zonal wind shear" throughout the manuscript.

- L49: Missing a link to WAM. The change of subject is too sudden and does not flow from the previous paragraph. Consider opening the section with lines 53-54

We have taken note of this missing link. We have replaced that statement with "One major atmospheric disturbance that contributes to the WAM is the presence of Mesoscale Convective Systems (MCSs) which supply around 30-80 % of the total rainfall during the WAM (Klein et al. 2018)".

- L94: "large-scale patterns" - Too vague. You should name the parameter used for the classification here.

Parameters used to represent large-scale patterns have been stated to clarify the statement on L94.

- L121: "SWA domain" - This domain should either be specified in latitude and longitude boundaries or displayed in a figure earlier on. Possibly both.

The domain of SWA has been shown earlier in Figure 2 with the latitude and longitude boundaries clearly specified in the figure caption.

- L124: This section requires more detail. For instance, what is the SOM topology? It would be useful to add a neighbor distances map and to evaluate SOM errors. The number of members in each cluster should also be given, preferably in Fig 1.

This has been taken into consideration and changes have been made to the section to capture the above suggestion. Based on generated Sammon maps, we were able to detect any error in the SOM easily. These Sammon maps use a non-linear mapping technique to create a two-dimensional image of the reference vectors where the distance between node vectors approximates the Euclidean distances in data space. We obtain a very ordered Sammon map in the training of the SOM, which made for a robust interpretation of nodal relationships. Similar types of circulation are close to each other in the SOM space and dissimilar circulations are furthest from each other, which is a characteristic of self-organizing maps.

- L128-129: This statement is true for many optional classification inputs. In the present study, the focus is on the tropics where geostrophic balance is not obvious, as seen by your results. Therefore, the choice to classify patterns using geopotential heights should be justified.

Different training variables were used to capture the regional atmospheric circulation. Our choice of the geopotential height at the low level was based on its ability to represent the impact of the West African Heat Low on the WAM cycle and also to identify the seasonal monsoon synoptic states over West Africa.

- L131: Each method has its advantages and disadvantages, and each can be more suited for a different study. Refrain from making conclusive statements.

We agree with the reviewer's view on refraining from making conclusive statements. In this line, we were merely reiterating what the literature says, however, we have modified this in the manuscript.

- L132: "data is not discretized and orthogonality is not forced" – Again, these are not clear advantages. The SOM's strengths and weaknesses should be discussed in the context of the present study.

The SOM strengths and weaknesses are discussed briefly in our study and the references given are for further reading. Indeed these strengths are well documented and are clear advantages to methods such as PCA and K-means clustering.

- L142: More information is required on what led to the choice of 9 clusters. Have you evaluated the network errors under the different configurations (SOM size and other parameters) to show that 9 is the most compatible?

The network errors are under different configurations for different sizes (4x4, 3x4, 3x3, and 2x3). On testing various sizes, a 9-node SOM was selected that adequately picks out the seasonal variation of rainfall over the region of study. The 2x3 resulted in a more generalized circulation archetype whiles the 3x3 represented a wider range of circulations with fewer redundancies.

- L153: Why not compare to non-MCS days within the node? This may highlight the signal you are after.

Thank you for the suggestion. This is well agreed but the main focus of the manuscript was to understand the synoptic state of the environment on MCS days. A look at non-MCS days can be done as future work to elaborate on signals.

- L154: regarding the T-test – on Which confidence level was it conducted? have you used any method to detect false positives in the multi-gridded test? See Wilks 2016 for example. Wilks, D.: "The stippling shows statistically significant grid points": How research results are routinely overstated and over-interpreted, and what to do about it, B. Am. Meteorol. Soc., 97, 2263–2273, https://doi.org/10.1175/BAMS-D-15-00267.1, 2016.

The T-test conducted was on a 95% confidence level. Anomaly plots highlight only regions at this confidence level (0.05 significance level) as well as wind vectors.

- L165: Why is the complete SOM not shown? This is not clear. If you choose to discard nodes altogether, you should show the full SOM map (9 nodes) first, then explain why not all nodes are relevant, and which ones were removed. The resulting 6-node SOM map should be shown in the context of the full SOM map, as the node locations on the SOM map are crucial for the SOM interpretation. This also raises the question: are the panels in Figures 2-9 arranged correctly? i.e., are neighboring nodes in these Figures also neighbors in the full SOM map? I suggest repeating the analysis for 6 nodes if that's what you end up analyzing, while completely removing irrelevant dates from the SOM input.

The complete SOM has been shown in an attached supplementary material (Figure S1) with the monthly distribution of node cases. Similar nodes from the 9-node case were combined in attaining the 2 x 3 nodes. We grouped nodes (1, 4, 7), and (6, 9) and kept 2, 3, 5, and 8 separate. That would also give 6 groupings seemingly representative of pre- and post-monsoon (1, 4, 7), peak-monsoon (6, 9), pre-monsoon only but different patterns (2 and 3), and post-monsoon only but different patterns (5 and 8). Nodes 1, 4, and 7 were considered as having out-of-monsoon conditions, with MCSs more likely far south. Nodes 6 and 9 show somewhat more Sahelian conditions while the tendency for monsoon retreat conditions was evident in nodes 5 and 8. We also observed pre-onset conditions in node 2 and in weaker terms in node 3. We have added some additional text to provide clarity for readers in Section 3.1.

[Figure]

Figure S1. The 3 x 3 SOM using daily ERA5 geopotential height Z at 925 hPa for Western Africa for the period 1981–2020. Insert is the monthly distribution of node cases based on the 3 X 3 SOM analysis

- L167-168: Even if some nodes are ignored, the numbering of the nodes should be as in the full SOM analysis, to be consistent with the complete SOM map.

We made sure the arrangement is consistent with the complete SOM with each node following distribution as in the 9-node case.

- Figure 1: Add the total number of members in each node. Consider normalizing per year and not per month.

This suggestion is respectfully disagreed with, as the purpose of the paper is to develop an understanding of the characteristics of the WAM and its association with MCSs. One of the main characteristics of the WAM is seasonal variability, so normalizing per year will not reveal this variability associated with it. Again, it will be difficult to attain the respective monsoon conditions such as pre-, peak- and post-monsoon when normalized per year.

- Figure 2: Grey grid can be removed to improve visibility. Also, be consistent with X-label intervals. Clarify whether these are daily means or 12 UTC composite.

The grey grid has been removed to improve visibility and the x-label has been made consistent. It has been clarified that they are 12 UTC composite.

- Figure 3: The low correspondence between winds and geopotential heights in the tropical region raises the question: what is the value of classifying by geopotential if it's not indicative of the flow field? Why not directly classify the velocity/ wind-speed fields?

The choice of low-level geopotential height was made in this manuscript because we wanted to consider somehow the influence of the West African Heat Low (WAHL) in influencing instability. At low levels, the geopotential height well describes the strength of the WAHL (Lavaysse et al. 2009; Biasutti et al. 2009). The wind field considered here is the mid-level winds which are consistent with geopotential height in that the mid-level easterly winds follow the northward and southward movement of the heat low.

- L211-218: This subsection is too slim. Either remove it or expand it to get to a conclusion. At the present state, this subsection does not contribute to the main motivation of this study and possibly draws the reader's attention from the main storyline.

This subsection has been removed as suggested.

- Figure 6: This domain should be shown earlier when first presenting the SWA domain.

The domain of SWA has been shown earlier in Figure 2.

- L258: unclear. Why does high humidity lead to cooling?

Thank you for this comment. High humidity does not necessarily lead to cooling. High humidity is just the introduction of more water vapor, which can be in a warmer or cooler atmosphere. The statement on 'enhanced moisture' has therefore been omitted.

- L279: This point was given as a well-known fact in the introduction, so I don't see what is the novelty here.

This point has been removed from this section

- L290: "eastern patterns" – This is not evident in figure 6. This issue should be discussed

Corrected

- L140: "pure node analysis" – What do you mean by this?

This statement has been removed.

**Technical corrections**
- L25: which=that.

Changed

- L38: variabilities=variability.

Changed

- L42: "Change, 2014" is not a reference, or is missing from the reference list.

Thank you for pointing this out. We have corrected this reference by replacing it with "IPCC 2014", with its corresponding reference added accordingly as IPCC: Climate Change, 2014: Synthesis Report. Contribution of Working groups I, II, and III to the Fifth Assessment Report of the Intergovernmental Panel on Climate Change [Core Working Team, R.K. Pachauri and L.A. Meyer (eds)]. IPCC, Geneva, Switzerland, 151, 2014.

- L88: environments= parameters?

Replaced 'environments' with 'environmental parameters'

- L97: Is this a correct use of the word stratify? Seems confusing to me. How about grouped/ separated?

Changed. We replaced 'stratify for' with 'grouped into'

- L108: product= data source.

Replaced 'product' with 'data source'

- L126: daily=daily mean.

Added 'mean' to 'daily'

- L182: SOMs=SOM.

Removed 's' from 'SOMs'

- L200: "much more strengthened" – Rephrase. Consider "Intensified", "Increased" and so on.

Replaced with "intensified"

- L211: "A further" = Further.

Corrected

- L213: during=within.

Replaced

- Figure 4: The term "moving mean" seems more fitting.

Replaced

- L225: This second subtitle is redundant.

Based on the structuring of the results, the analysis has been grouped under various subtitles, of which the second subtitle well describes the analysis beneath it. We would therefore want to leave the subtitle as such.

- L244: Repetitive.

The statement on L244 has been removed

- Figure 7: The colors appear saturated. Expand the color map beyond 2K to avoid this.

The color map has been expanded for figure 7

- L274: observes= demonstrates/ exhibits.

Replaced observes with exhibits

- L290: observe = show/ depict.

Replaced observe with depict

- L308-311: Long sentence, consider splitting.

Thank you for this observation. The sentence has been split to read as follows: "Node 1 climatological conditions depict both, very low initial shear and TCWV. This illustrates the relatively storm-hostile mean conditions for this node, predominantly representing dry season conditions and explaining the low storm frequency of only 0.13 per day."

- L323: "making node 2…" This sentence is unclear, rephrase.

The sentence has been corrected by removing the last part.

- L330: "This season" – Unclear which season is that.

The sentence has been corrected to capture "this season" as "the monsoon season".

---

## Referee Report (RR1)

Review V2 for "Classification of Large-Scale Environments that drive the formation of Mesoscale Convective Systems over Southern West Africa" by Nkrumah et al.

**Overview**

Compared to the initial version, the authors were able to make some improvements on the paper. However, after having received the responses, I do not feel that every question I had was fully answered. Furthermore, the authors should overall consider transferring some more of the explanations they have given in response to my questions to the main text. I therefore recommend minor revision.

**General comments/questions on the responses**

- The authors argued in their response that the "scope of the work looked at investigating environmental conditions favourable for MCSs over the region". One aspect of it includes under which conditions MCSs might be triggered in the first place and further develop in southern West Africa. Therefore, it was a bit disappointing to see that there was no attempt to address some of the general comments I had (e.g CIN). For instance, it is known that MCSs can develop in high-CAPE/high-CIN situations where high CIN inhibits a premature initiation of smaller-scale convection that allows CAPE to further build up. Once CIN breaks down or is overcome, e.g. through moisture convergence or convergent motions at elevated terrain, vertical wind shear becomes relevant for the consequent evolution. While this has been observed for the midlatitudes and also partly for the Sahel, MCSs southern West Africa may be initiated differently in a moister environment. I do believe that this aspect is missing in the paper and is not beyond the scope. My suggestion: Have a look into anomalies of CIN and moisture (flux) convergence the same way as CAPE.

**Specific comments/questions on the responses**

- On the question why 925 hPa specific humidity was replaced by TCWV: *"As pointed out by the referee, we considered the TCWV due to its ability to represent the total gaseous water in the vertical column of the atmosphere which is influenced by the evolution of the humidity field. TCWV represents the precipitable water the atmosphere holds better than the humidity. We, therefore, had to show both since in the first instance (i.e. 925 hPa humidity) we were looking at an environment that is suitable for instabilities in the atmosphere, of which humidity forms a part."* I think this needs to be added to the text then since there was no motivation given of why TCWV was suddenly used in Fig. 10 and not elsewhere.

- On the question how the authors determined the rainfall amount: "*The rainfall amount was determined from rainfall snapshots of the "high-quality precipitation" (HQ) a field within the Integrated Multi-satellite Retrievals for Global Precipitation Measurement (IMERG; Huffman et al. 2019) dataset. This has been included in the manuscript as follows: This can include the same MCS at several timesteps in a day. Corresponding rainfall snapshots were sampled from the "high-quality precipitation" (HQ) field within the Integrated Multi-satellite Retrievals for Global Precipitation Measurement (IMERG; Huffman et al. 2019) dataset.*". Why did the authors use the HQ fields of IMERG only? If really variable "HQprecipitation" was used then the authors should have experienced large data gaps since PMW satellites alone cannot fully cover the region at a given timestep. Sure that the variable "precipitationCal" was not used instead?

- On why no MCS are seen in DJF in Fig. 5: "*The focus of MCSs over the study area in this study is during the rainfall season of the SWA domain which mainly starts in March and ends in November. February recorded zero because it wasn't considered in the frame of this work.*". Then Fig. 5 is misleading, and the x-axis should be truncated to the relevant months. Is this also the case for the numbers in Fig. 6? In any case, unless I missed it, this should be mentioned in the data section as well.

**Other specific comments/questions**

L219: 925 hPa is not exactly surface level.

L235: "The patterns demonstrate northward transport…". Of what?

L241: "Generally, the presence of zonal wind shear can be seen as a necessary condition…" For what?

Fig.4: How is it possible that the wind shear is negative? Is the directional information included as well, i.e. the direction of the shear vector? Then the authors need to provide more information on the sign of the wind shear.

Fig. 8: Following on the comment on Fig. 4 above, a negative anomaly in wind shear can have a different meaning when wind shear itself can be negative. Node 3 for instance, where the western Sahel exhibit a negative anomaly on climatologically negative wind shear values. What does that mean?

---

## Referee Report (RR2)

Revised WCD paper V2

The revised manuscript shows the full SOM analysis in supplementary material but does not sufficiently address the previously raised concerns. Specifically, major questions are still pending regarding the SOM classification, which lies at the heart of the analysis. The authors refrain from providing any metrics to estimate the SOM robustness, nor do I better understand why the original 9-node SOM is not used. The authors do not share my concern regarding the evidently low correspondence between low-level geopotential heights and winds in the region of interest. While the subject of the study is important, and relevant for the scope of the journal, substantial issues arise from the present methodology. I conclude that the manuscript should still undergo **major revisions before being accepted.**

**Major revisions:**

1) The subjective formation of a 6-node SOM from an objectively defined 9-node SOM raises issues. The classified geopotential patterns' robustness (or confidence level) is unclear. It is important to account for each node's internal variability and to mark which regions of the flow are significant. From previous experience, the statistical significance of the SOM nodes may not include the entire domain, and it is important to clarify the regions where the pattern is indeed robust. This is helpful to determine the relevance of the mean node states displayed in figure 10, i.e., what is the standard deviation of the mean environmental conditions? If they largely overlap with that of MCS days, then the separation may not be as meaningful as suggested by the authors. Furthermore, no topological or quantization errors are displayed, nor is a Sammon map, making assessing the SOM's performance virtually impossible. Also, the relative frequency of each node is not given, though it is clear that the distribution is far from equal. This leads to peak Monsoon node 6 showing a reduced MCS frequency despite containing the largest amount of MCSs. This is problematic since the main frequency of each node is unclear. I.e., if node 6 is twice (or more) as frequent as node 5, it is reasonable to expect a lower MCS frequency that may be an attribute of inner cluster variability (or noise) rather than a dynamic feature.

2) There is a problem with the logic of the present analysis: Nodes are associated with monsoon phases mostly based on their seasonality. MCSs make up 30-80% of the monsoon precipitation, so they are clearly more common during peak monsoon stages. This suggests that pre-monsoon nodes will have significantly fewer MCSs and hence their response to MCS appears larger. As a result, seen in Figure 10, only node 1 seem to show a significant response to MCS, while for the other nodes, the mean state is well within the STD of the MCS conditions. This undermines the main point of the paper.

3) The MCS data should be better treated and displayed. As innovative as combining satellite data to such a large-scale perspective is, it eventually ends up only as a cloud indicating all MCSs within each node. It is worth visualizing the data in a manner that will ease the interpretation of the results. For instance, it is unknown what the MCS spatial distribution actually looks like for each node. Overlapping dots are invisible and may hide preferable locations for MCSs. The mean number of MCSs per MCS day per node is not given, thus it is possible for example that while MCS "daily" frequency peaks on node 5, the overall MCS frequency may be larger for node 6, simply by having more MCSs per MCS day. Such discrepancies in the MCS data should be addressed and studied within the present manuscript, including spatial variability.

4) Information on the SOM dynamics also seems relevant, e.g., how long does each node persist? are the nodes typically changing on a time scale of days or weeks? Which nodal transitions are frequent, which are rare, and how does this relate to the dynamical interpretation of the SOM as indicating the monsoon phases? Such information is valuable to evaluate the SOM's consistency in mapping consecutive days, and support the derived conclusions.

**Minor comments:**

L140: the heat low is not well captured relative to similar SOM analysis. I've commented more about that above.

L144: ts = typo?

L156: Q and T errors should be presented per cluster

L160: what led to the choice of 9 members? Was it purely a qualitative choice?

L210: Why is a heat low evident as a high-pressure area?

L212: "and linked southward retreat" – rephrase

L238: "weaker geopotential heights representative of high-pressure areas" – unclear. Rephrase.

L335: the use of the term "storm" is confusing. Stick to MCS.

L356 & 357: use "show/shown" instead of the passive "observe/observed".

L359: remove double dots

L364: "with frequent convective activities during peak monsoon" - Why is the same not true for node 5 with the most frequent MCS days?

L376: "… states and then examined…" = … states and examined…

---

## Referee Report (RR3)

Review V3 for "Classification of Large-Scale Environments that drive the formation of Mesoscale Convective Systems over Southern West Africa" by Nkrumah et al.

**Overview**

Overall, the latest iteration of the present manuscript has included an extended set of nodes, which, in my view, has improved the coherence of the paper. From my side, there are only minor revisions.

**General comments/questions on the responses**

- The aspect of different within-season states is an interesting one as they can transition across each other within days. Therefore, can the authors further, but briefly, elaborate on potential dynamical sources of these transitions? From Fig. 4, it appears that low to midlevel westerlies are more pronounced in the top row compared to the rest, which might show the impact of extratropical signals. Overall, these transitions seem to have an impact on the probability of MCS occurrence (Fig. 8), which warrants at least a short evaluation.

**Other specific comments/questions**

L135:     "TCWV represents the precipitable water the atmosphere holds better than the humidity." I do not get this sentence.

L140:     "km2". Set the "2" in superscript.

L182:     "Based on 6 different large-scale node patterns …" Should be nine!?

Fig. 3:   Colours + patterns for the within-season nodes are visually not necessarily well distinguishable. It helps though that the bars are ordered the same way as in the legend.

Fig. 8:   It appears to me that signals in node 1 are dominated by land-sea breeze convection along the coast which are gradually suppressed in node 4 and 7. Therefore, the large-scale settings seemingly facilitate such rather local-scale developments. Maybe the authors can briefly pick up on this in the text.

L359:     "… reveal a widespread increase in zonal wind shear anomaly…". But in this case, it means (mostly) a weaker westerly shear? The authors may work with directional indications for clarity.

---

## Referee Report (RR4)

Revised WCD paper V3

I find the revised manuscript as a significant improvement. The full SOM appears to better grasp the variability in the system, as is evident by the sharp clustering in Fig. 13. The persistency-transition analysis and the spatial variability of MCSs greatly enhance the scope of the paper, and lead to valuable findings while enabling a deeper understanding of the SOM analysis.

I think the manuscript should be accepted for publication following several minor comments:

**Minor comments:**

L1: Be consistent with capital letters.

L21: I suggest adding a motivational sentence stating the importance of the topic.

L35: … the SOM identified…

L145: The sentence is incomplete.

L174: analysis

L175: remove "to choose"

L316: MCSs rarely develop…

L317: address the similar frequencies of clusters 1 and 9.

L383: I would replace "swath" with a more common word.

L439: arrange the spacing.

L469: southward

L471: why "presumably"? Is this not shown in Fig. 9?

L491-492: Why then don't these nodes show maximum MCS frequencies?

Well done!

---

## Author Response (AR2)

**RESPONSE TO REFEREES**

**Referee 1:** Review V2 for "Classification of Large-Scale Environments that drive the formation of Mesoscale Convective Systems over Southern West Africa" by Nkrumah et al.
**Overview:** Compared to the initial version, the authors were able to make some improvements on the paper. However, after having received the responses, I do not feel that every question I had was fully answered. Furthermore, the authors should overall consider transferring some more of the explanations they have given in response to my questions to the main text. I therefore recommend minor revision.

Thank you for the in-depth comments and specific analysis advice the referee provided for our manuscript. This helped significantly improve our results' robustness and sharpen our discussion. Below, we give a point-by-point response to the comments with line references to the new manuscript version. We also provide a tracked changes document highlighting the differences from the previous version.

• **General comments/questions on the responses**
  ● The authors argued in their response that the "scope of the work looked at investigating environmental conditions favorable for MCSs over the region". One aspect of it includes under which conditions MCSs might be triggered in the first place and further develop in southern West Africa. Therefore, it was a bit disappointing to see that there was no attempt to address some of the general comments I had (e.g CIN). For instance, it is known that MCSs can develop in high-CAPE/high-CIN situations where high CIN inhibits a premature initiation of smaller-scale convection that allows CAPE to further build up. Once CIN breaks down or is overcome, e.g. through moisture convergence or convergent motions at elevated terrain, vertical wind shear becomes relevant for the consequent evolution. While this has been observed for the midlatitudes and also partly for the Sahel, MCSs southern West Africa may be initiated differently in a moister environment. I do believe that this aspect is missing in the paper and is not beyond the scope. My suggestion: Have a look into anomalies of CIN and moisture (flux) convergence the same way as CAPE.

Thank you for this comment, it is, of course, correct that anomalies of certain variables will be more strongly linked to what allows MCSs to be maintained and organized (as the reviewer said, wind shear) and what may allow or inhibit their initiation (as CIN). We cannot disentangle these different life stages here as we're not looking at MCS tracks, but we now include anomalies of CIN in the supplementary as suggested by the reviewer, finding that all nodes show decreased CIN over SWA on MCS days. This is included in the main text in section 4.2:

" *Overall, all nodes show positive CAPE and negative convective inhibition (c.f. Supplementary Fig. S4) anomalies for MCS days in parts of SWA, creating an environment sufficiently unstable to support the development of convection.*"

**Specific comments/questions on the responses**
  ● On the question why 925 hPa specific humidity was replaced by TCWV: "As pointed out by the referee, we considered the TCWV due to its ability to represent the total gaseous water in the vertical column of the atmosphere which is influenced by the evolution of the humidity field. TCWV represents the precipitable water the atmosphere holds better than the humidity. We, therefore, had to show both since in the first instance (i.e. 925 hPa humidity) we were looking at an environment that is suitable for instabilities in the atmosphere, of which humidity forms a part." I think this needs to be added to the text then since there was no motivation given of why TCWV was suddenly used in Fig. 10 and not elsewhere.

Thank you for pointing this out. We now specify the link between low-level moisture anomalies and instability changes in Section 4.2:

*"Overall, all nodes show positive CAPE and negative convective inhibition (c.f. Supplementary Fig. S4) anomalies for MCS days in parts of SWA, creating an environment sufficiently unstable to support the development of convection. The close alignment with regions of increased low-level humidity (Fig. 9) suggests increased low-level moisture advection as the main driver for these instability changes."*

and added an introductory sentence as justification for the use of TCWV in Section 4.3 as follows:

*"TCWV instead of single-level specific humidity is used here to capture the changes in total moisture available to MCSs under the different regimes."*

- On the question how the authors determined the rainfall amount: "The rainfall amount was determined from rainfall snapshots of the ''high-quality precipitation'' (HQ) a field within the Integrated Multi-satellite Retrievals for Global Precipitation Measurement (IMERG; Huffman et al. 2019) dataset. This has been included in the manuscript as follows: This can include the same MCS at several timesteps in a day. Corresponding rainfall snapshots were sampled from the ''high-quality precipitation'' (HQ) field within the Integrated Multi-satellite Retrievals for Global Precipitation Measurement (IMERG; Huffman et al. 2019) dataset.". Why did the authors use the HQ fields of IMERG only? If really variable "HQprecipitation" was used then the authors should have experienced large data gaps since PMW satellites alone cannot fully cover the region at a given timestep. Sure that the variable "precipitationCal" was not used instead?

Thank you for catching this. In previous studies, we indeed used the PMW product to match up cloud top temperature with rainfall, as for applications where it is not crucial to capture all MCS cases (the reviewer correctly points out that HQprecipitation only covers swaths), the PMW rainfall better conserves the spatial structure of the rainfall field and thus provides a cleaner cloud top temperature/precipitation field relationship. However, given that in this work we only use the maximum rainfall rather than its spatial structure within the MCS cloud envelope, we used the spatially continuous "precipitationCal" product to capture as many raining MCSs as possible. The full paragraph will be included in the manuscript as follows:

*"Twelve years of MCS snapshots (2004–15) detected from Meteosat Second Generation 10.8 μm-band brightness temperatures (Schmetz et al., 2002, EUMETSAT 2021) are used to define MCS days in this study. Following (Klein et al., 2021), an MCS is defined here as a -50ºC contiguous cloud area larger than 5000 km². We consider the MCS images every half hour, for which they are matched up with the half-hourly Integrated Multi-satellite Retrievals for Global Precipitation Measurement (IMERG; Huffman et al. 2019) dataset, using the merged microwave / infra-red ("precipitationCal") rainfall product. An "MCS day" is then defined as a day with at least one hour containing 5 simultaneously existing MCSs between 16 and 1900 UTC with maximum rainfall >5mm within the SWA domain."*

- On why no MCS are seen in DJF in Fig. 5: "The focus of MCSs over the study area in this study is during the rainfall season of the SWA domain which mainly starts in March and ends in November. February recorded zero because it wasn't considered in the frame of this work.". Then Fig. 5 is misleading, and the x-axis should be truncated to the relevant months. Is this also the case for the numbers in Fig. 6? In any case, unless I missed it, this should be mentioned in the data section as well.

We apologize for the confusion as to which months were included in the analysis. For completeness, we now processed and included the MCSs for the full annual cycle (Jan-Dec), which is reflected in Figs. 1, 7, and Fig. 8, and all node anomalies with MCS days in DJF.

**Other specific comments/questions**
- L219: 925 hPa is not exactly surface level.

The statement has been rewritten to reflect the 925 hPa level.

- L235: "The patterns demonstrate northward transport...". Of what?

The statement has been rewritten to read "The patterns in zonal wind shear from first to third column nodes illustrate a strong link of high-shear areas to the propagation of the WAM cycle, and these areas widen as the zonal shear band moves further inland."

- L241: "Generally, the presence of zonal wind shear can be seen as a necessary condition..." For What?

This statement has been removed.

- Fig.4: How is it possible that the wind shear is negative? Is the directional information included as well, i.e. the direction of the shear vector? Then the authors need to provide more information on the sign of the wind shear.

As outlined in section 2.1., the shear indicator used here corresponds to the zonal wind difference between 925hPa and 600 hPa, which is the main direction in which MCSs propagate (east to west). We thus present enhanced easterly shear as positive anomalies and follow the convention that easterly shear and easterly anomalies are presented with a positive sign across all plots, as these are positively related to storm development. This is now explained in the methodological data section as follows:

*"Due to the main direction in which MCSs propagate across the SWA region (east to west), enhanced easterly zonal wind shear anomalies are presented as positive anomalies as these are positively related to storm development."*

- Fig. 8: Following on the comment on Fig. 4 above, a negative anomaly in wind shear can have a different meaning when wind shear itself can be negative. Node 3 for instance, where the western Sahel exhibit a negative anomaly on climatologically negative wind shear values. What does that mean?

This means a strengthening of westerly shear, which is not favoring MCS development or maintenance.

**Referee 2:**
- The revised manuscript shows the full SOM analysis in supplementary material but does not sufficiently address the previously raised concerns. Specifically, major questions are still pending regarding the SOM classification, which lies at the heart of the analysis. The authors refrain from providing any metrics to estimate the SOM robustness, nor do I better understand why the original 9-node SOM is not used.

Thank you for your advice. Regarding the SOM classification, we evaluate the quantization error. An optimal SOM is obtained when the average Euclidean distance is the minimum (the quantization error is the smallest). Once the average quantization error has been minimized, the relationships between the predictor and node data are investigated. Based on evaluating the quantization errors for node sizes 2x3, 3x3, and 3x4, it was observed that node size 3x3 had the smallest quantization error. On testing various sizes, a 9-node SOM was selected that adequately picks out the seasonal variation of rainfall over the region of study. The 2x3 resulted in a more generalized circulation archetype while the 3x3 represented a wider range of circulations with fewer redundancies.

[Figure]

**Figure S1.** Total quantization error (gpm) of all grid points in the geopotential height patterns mapped to each SOM node.

We have attached here figures of calculated quantization errors for node sizes 2x3, 3x3, and 3x4. This figure has been added to the supplementary material as Figure S1.

Following this comment, we now provide the original 9-node SOM information in the manuscript.

- The authors do not share my concern regarding the evidently low correspondence between low-level geopotential heights and winds in the region of interest.

We assume the reviewer refers to Fig. 3 (old manuscript, Fig. 5 in new manuscript), which however shows geopotential height at 925 hPa (the field the SOM analysis is based on) and flow vectors at 650 hPa (the MCS steering wind level in the region), where consequently the fields do not evidently correspond.  For illustration, please refer to the plots below (Figs. S2 and S3) where 925 hPa and 650 hPa geopotential height and wind field are plotted together for the same pressure levels. These are now also available in the supplementary material. This is now better clarified in the text:

*"The SOM classification of different synoptic states was based on 925 hPa geopotential heights, with resulting patterns shown in Fig. 4. [...] The overlaid 650 hPa wind field reveals mean easterly wind conditions at MCS steering levels across all nodes, suggesting that the dominant propagation direction for MCSs remains east to west for all synoptic states."*

[Figure]

**Figure S2.** 12 UTC composites of 925-hPa geopotential height (shading; gpm) and 925-hPa winds (vectors; m s$^{-1}$ ) in 9 nodes based on SOM analysis. The purple box depicts the SWA region (5º–9ºN, 10ºW–10ºE)

[Figure]

**Figure S3.** 12 UTC composites of 650-hPa geopotential height (shading; gpm) and 650-hPa winds (vectors; m s$^{-1}$ ) in 9 nodes based on SOM analysis. The purple box depicts the SWA region (5°–9°N, 10°W–10°E)

- While the subject of the study is important and relevant for the scope of the journal, substantial issues arise from the present methodology. I conclude that the manuscript should still undergo major revisions before being accepted.

**Major revisions:**
- The subjective formation of a 6-node SOM from an objectively defined 9-node SOM raises issues. The classified geopotential patterns' robustness (or confidence level) is unclear. It is important to account for each node's internal variability and to mark which regions of the flow are significant.

Thank you very much for this comment, we now show the original 9-node SOM for the manuscript which, while creating some redundancies, also allowed us to discuss the node matrix structure in more detail, as illustrated by new Figures 2 and 3 looking at node persistence and node-to-node transitions, respectively, as later suggested by the reviewer.

Regarding gph pattern robustness, we evaluated the quality and/or robustness of the SOM in this study based on the learning quality indicator, which is determined through the measurement of the quantization error (QE). Based on evaluating the quantization errors for node sizes 2x3, 3x3, and 3x4, we found that node size 3x3 showed a quantization error minimum compared to the higher/lower node number cases. On testing these various sizes, we hence selected the 9-node SOM as it also adequately picks out the seasonal variation of circulation over the region

of study. The 2x3 resulted in a more generalized circulation archetype while the 3x3 represented a wider range of circulations with fewer redundancies. This is now detailed in the SOM Methods section: "*Based on SOM_PAK, we tested node sizes 2x3, 3x3, and 3x4, using the quantization error (QE) as an indicator for quality and robustness of the respective node size. We find a minimized QE for 3x3 (c.f. Supplementary Figure S1), which, from visual inspection, also shows a larger number of distinct circulation features than 2x3 while producing fewer redundancies than 3x4. Thus, all following analyses are based on the 3x3 node matrix.*"

- From previous experience, the statistical significance of the SOM nodes may not include the entire domain, and it is important to clarify the regions where the pattern is indeed robust.

Thank you, it is true that the node presentation on the large West African domain does not necessarily imply statistically significant differences between nodes for each part of the domain or on a pixel basis. As outlined above, the SOM nodes were identified based on synoptic-scale patterns and we now provide the QE errors. Unfortunately, the reviewer has not provided a test strategy or particular test they would like to see performed for further 'node significance' testing. We hope the information we now provide on node persistence and node-to-node transition demonstrates node differences more satisfactorily.

To explore within-node variability, we also performed a node-mean bootstrapping with 5000 resamples per node and per domain pixel, with Fig. S4 below representing the width of the 95% confidence interval for average geopotential height maps (testing representativeness of geopotential means as presented in the revised manuscript in Fig. 4). We find that for all tested ERA5 pixels, the calculated means as shown in Fig. 4 lie within the respective 95% confidence interval. As visible from Fig. S4, per-pixel 95% confidence intervals are generally small in comparison to large-scale geopotential height gradients and decrease further towards the SWA domain (red shading).

[Figure]

**Figure S4.** 12 UTC 925-hPa geopotential height 95% confidence interval widths (shading; gpm) based on bootstrapping with 5000 resamples for the 9 node day groups based on the SOM analysis.

- This is helpful to determine the relevance of the mean node states displayed in figure 10, i.e., what is the standard deviation of the mean environmental conditions? If they largely overlap with that of MCS days, then the separation may not be as meaningful as suggested by the authors.

In Figure 10 (Fig.13 in updated manuscript), instead of adding standard deviation whiskers to node mean states and MCS-day states (which creates visualization problems), we now provide the standard deviations of the node mean

states and perform Welch's t-tests between node mean states and MCS-day states to indicate for which nodes the mean MCS-day states are significantly different in TCWV and shear. As expected, this highlights that certain nodes are favorable for MCSs in SWA as per their mean state (e.g. node 3, which does not show a significant difference in either TCWV or shear and thus suggests the most favorable mean synoptic conditions for MCSs), while other nodes show significant within-node differences for MCS days only in shear (e.g. peak monsoon nodes 6,9) or both TCWV and shear.

- Furthermore, no topological or quantization errors are displayed, nor is a Sammon map, making assessing the SOM's performance virtually impossible.

Thank you for raising this point.
Here we display the quantization errors for the different SOM sizes we tested. Based on evaluating the quantization errors for node sizes 2x3, 3x3, and 3x4, it was observed that node size 3x3 had the smallest quantization error. On testing various sizes, a 9-node SOM was selected that adequately picks out the seasonal variation of rainfall over the region of study. The 2x3 resulted in a more generalized circulation archetype whiles the 3x3 represented a wider range of circulations with fewer redundancies.

We have included a figure (**Fig. S1**) of calculated quantization errors for node sizes 2x3, 3x3, and 3x4 in the supplementary material.

- Also, the relative frequency of each node is not given, though it is clear that the distribution is far from equal. This leads to peak Monsoon node 6 showing a reduced MCS frequency despite containing the largest amount of MCSs. This is problematic since the main frequency of each node is unclear. I.e., if node 6 is twice (or more) as frequent as node 5, it is reasonable to expect a lower MCS frequency that may be an attribute of inner cluster variability (or noise) rather than a dynamic feature.

Thank you for your comment, the absolute and relative frequency of each node is now given in the title of Fig. 1, alongside the node cases per month. We wouldn't consider the fact that (previous) monsoon node 6 showed reduced MCS frequency per day despite containing the largest number of MCSs (solely due to it containing the largest number of node days) as problematic. The metric MCS frequency/day implicitly contains the number of days per node and answers the question of whether a particular node is more or less favorable for MCSs in SWA on a particular day, as intended. Please note that in SWA, the "peak monsoon" period includes a dip in MCS frequencies in southern West Africa, including the little dry season, when MCS frequency is in fact lower, as shown in Fig. 7. We now revised our node classification into "dry season, transition season and monsoon season" and discuss the relationship of MCS frequencies/day with the presented MCS annual cycle in more detail in the respective first paragraph of section 4.2:
"*The location of MCSs during node days is represented in Fig. 8. Comparing daily MCS frequencies, we find that MCSs are most likely to develop under transition node (2,5,8) conditions (2.8 MCSs per day) featuring a northward-displaced moisture anomaly (Fig. 9). Given the transition nodes occur predominantly during pre-monsoon (late March to June) and post-monsoon (from September to November) - the major and the minor rainy season respectively in SWA (cf. Fig.~1), these patterns may in some cases be representative of early monsoon onset and a delayed monsoon retreat respectively. MCSs seldom develop under dry node (1,4,7) conditions, with as low as 0.6 MCSs per day*"

- There is a problem with the logic of the present analysis: Nodes are associated with monsoon phases mostly based on their seasonality. MCSs make up 30-80% of the monsoon precipitation, so they are clearly more common during peak monsoon stages.

We apologize that the given information in the manuscript seems to give rise to this misunderstanding, but for southern West Africa, MCSs are not the most common during peak monsoon stages, as Fig.5 (Fig. 7 in the revised manuscript) illustrates. Overall MCS numbers in southern West Africa are lower during peak monsoon months July and August, due to the bulk monsoon band moving further inland during the monsoon peak, creating drier conditions in southern and coastal regions. Large parts of the evaluated region have a bimodal rainfall regime during pre/early and post/late monsoon stages, created by the northward passing and subsequently southward passing monsoonal rain band. Depending on exact locations, MCS frequency peaks thus lie in pre-/and post-monsoon phases (now termed "transition season"). This is reflected in the MCS/day metric. The "peak monsoon" expression is not to be used synonymously with peak rainfall or peak MCS occurrence but reflects the monsoon movement stage. This has been added to the introduction section to read as follows:

"*The SWA region is different from its Sahelian counterpart in its closer proximity to the ocean and a distinct bimodal rainfall seasonality. The WAM stages can broadly be classified into a dry season when north-easterly Harmattan winds prevail over most of West Africa during December-February when rainfall mostly occurs off the southern coast of the continent (Thorncroft et al 2011), and the monsoon season from July-September, initiated by a striking jump of the monsoonal rainfall band from coastal regions to the Sahel (Hagos and Cook, 2007). The monsoon months thus represent the unimodal Sahelian rainfall season. In SWA however, the majority of rainfall occurs between the dry months and monsoon months, when the monsoon rainband first passes northward over southern regions from March to June and subsequently moves southward again when the monsoon retreats in October (e.g. Maranan et al 2018, Klein et al 2021). Here, we summarize these months when SWA receives most of its rainfall as transition season.*"

- This suggests that pre-monsoon nodes will have significantly fewer MCSs and hence their response to MCS appears larger. As a result, seen in Figure 10, only node 1 seem to show a significant response to MCS, while for the other nodes, the mean state is well within the STD of the MCS conditions. This undermines the main point of the paper.

Thank you for this comment. We first want to point out that in our manuscript, we are transparent about the differences in MCS numbers per node as per the given MCS/day frequencies (old Fig. 4, now Fig. 8), which exactly addresses the question as to what within-node conditions would allow MCS development. We additionally now added absolute MCS numbers per node to Fig. 1 in the updated manuscript.

We would argue that illustrating that certain nodes are MCS-favorable in their mean state while others are not (i.e. show large anomalies in Section 4.2) is exactly one of the important main points of the paper, rather than undermining any point made, i.e. no or little difference between mean node state and MCS-day state identifies nodes (large- scale synoptic states) that are on average MCS favorable in southern West Africa. However, as our updated Fig. 13 illustrates, only node 3 shows insignificant changes in both TCWV and shear for the MCS day mean compared to the node mean. Higher or lower MCS frequencies within a node are not a problem of the analysis but a consequence of how frequently respective SOMS-defined synoptic conditions allow the development of convection in SWA (producing the MCS-day anomalies shown in Fig 13 in the updated manuscript). It is thus perhaps not an entirely unexpected result that nodes that represent on average dry season conditions (e.g. node 7) "need" larger increases in TCWV and shear to allow MCS development.
In addition to updating Figure 13, we now clarify that nodes that display the largest changes needed for MCS development are nodes that exhibit on average atmospheric conditions most hostile to convection and thus feature the smallest total within-node MCS numbers as part of our revised discussion & conclusion section.

- The MCS data should be better treated and displayed. As innovative as combining satellite data to such a large-scale perspective is, it eventually ends up only as a cloud indicating all MCSs within each node. It is

worth visualizing the data in a manner that will ease the interpretation of the results. For instance, it is unknown what the MCS spatial distribution actually looks like for each node. Overlapping dots are invisible and may hide preferable locations for MCSs. The mean number of MCSs per MCS day per node is not given, thus it is possible for example that while MCS "daily" frequency peaks on node 5, the overall MCS frequency may be larger for node 6, simply by having more MCSs per MCS day. Such discrepancies in the MCS data should be addressed and studied within the present manuscript, including spatial variability.

The MCS/day metric was calculated as the absolute number of detected MCSs at 1800 divided by the number of node days. If more than one MCS exists at that time on any single day, this frequency contribution is thus taken into account (I.e. the number does not merely reflect the number of MCS days per node). As per the explanation of the seasonal cycle of MCSs in southern West Africa given above, our result of a higher MCS frequency for node 5 in the previous version of the manuscript was reasonable given that node 5 predominantly covered September and October days, the peak of the second rainfall season in southern West Africa (these nodes are now termed "transition nodes"). We added a new figure showing a density plot of MCSs (rather than individual points) with a zoomed-in domain to facilitate pattern evaluation and provide the needed detail on how the MCS frequency/day is calculated in the caption of that figure (Fig. 8 in the revised manuscript). We also include a description of the typical annual cycle and the relations of the rainy season to the West African monsoon progression to the introduction as mentioned above.

- Information on the SOM dynamics also seems relevant, e.g., how long does each node persist? are the nodes typically changing on a time scale of days or weeks? Which nodal transitions are frequent, which are rare, and how does this relate to the dynamical interpretation of the SOM as indicating the monsoon phases? Such information is valuable to evaluate the SOM's consistency in mapping consecutive days and support the derived conclusions.

Thank you for this question which helped us better explore the node differences - and made the presentation of the full 3x3 SOM matrix very useful. We now added a figure showing typical (and clearly different!) node persistence as well as node transitions in the manuscript as Fig. 3 and Fig. 4, which includes an added new section on node persistence and node-to-node transitions, including in the entirely revised discussion part.

**Minor comments:**
L140: the heat low is not well captured relative to similar SOM analysis. I've commented more about that above.
See provided plot that illustrates consistency above.

L144: ts = typo?
Yes it is a typo. This has been corrected

L156: Q and T errors should be presented per cluster
The errors are now shown in Fig. S1 in the supplementary material.

L160: what led to the choice of 9 members? Was it purely a qualitative choice?

On testing various sizes, a 9-node SOM was chosen that adequately picks out the seasonal variation of rainfall over the region of study. The 2x3 resulted in a more generalized circulation archetype whiles the 3x3 represented a wider range of circulations with fewer redundancies. This was based on evaluating the quantization errors for node sizes 2x3, 3x3, and 3x4, which saw that node size 3x3 had the smallest quantization error. The choice was therefore based

L210: Why is a heat low evident as a high-pressure area?

As is typical for thermal lows, the SHL exhibits low pressure for near-surface levels (the 925hPa level at which the geopotential height is shown), which turns into a high pressure circulation above at 650hPa for which the wind pattern is shown. To avoid any more confusion, we now show both, the 925hpa geopotential with circulation field, and 650hPa with circulation field here and in the supplementary materials – illustrating correspondence between geopotential height and flow, and only refer to the low pressure system in the low-levels that is known as the Saharan Heat Low.

L212: "and linked southward retreat" – rephrase
This statement has been removed.

L238: "weaker geopotential heights representative of high-pressure areas" – unclear. Rephrase.
This statement has been removed.

L335: the use of the term "storm" is confusing. Stick to MCS.
The term "storm" has been changed to "MCS".

L356 & 357: use "show/shown" instead of the passive "observe/observed".
These changes have been done

L359: remove double dots
Double dots have been removed

L364: "with frequent convective activities during peak monsoon" - Why is the same not true for node 5 with the most frequent MCS days?
This has been revised, now considering the full 9 node structure.

L376: "..states and then examined.." - .. "states and examined..."
 Changed.

---

## Author Response (AR3)

**RESPONSE TO REFEREES**

**Referee #1:**
Review V3 for "Classification of Large-Scale Environments that drive the formation of Mesoscale Convective Systems over Southern West Africa" by Nkrumah et al.

**Overview:** Overall, the latest iteration of the present manuscript has included an extended set of nodes, which, in my view, has improved the coherence of the paper. From my side, there are only minor revisions.

We are grateful for the in-depth constructive review the referee provided for our manuscript. This has helped significantly to improve our results' robustness and sharpen our discussion.

**General comments/questions on the responses**
The aspect of different within-season states is an interesting one as they can transition across each other within days. Therefore, can the authors further, but briefly, elaborate on the potential dynamical sources of these transitions? From Fig. 4, it appears that low to midlevel westerlies are more pronounced in the top row compared to the rest, which might show the impact of extratropical signals. Overall, these transitions seem to have an impact on the probability of MCS occurrence (Fig. 8), which warrants at least a short evaluation.

Thank you for this comment, it is correct that this may be linked to extratropical wave passage, similarly however (as is particularly visible for node 3) this change in the mid-level westerlies may also or in addition be linked to fluctuations of the West African Heat Low, as was described by Lavaysse et al. 2009, which was shown to take place on the order of days, in some cases modified by dust concentration (Lavaysse et al. 2011). This is now included in the main text in page 9 as follows:

"As was shown in Fig. 3, the discussed node states have an average duration on the order of days, indicating frequent transitions. Notably, mid-level westerlies are strengthened or shifted southwards for all top-row nodes in Fig. 4, which is associated with increased probability for MCS occurrence compared to other nodes, as we will outline later (c.f. Fig. 8). Potential synoptic factors that may drive the frequent node transitions and hence affect MCS frequency include extratropical waves, as well as the WAHL that is most pronounced for top-row nodes. WAHL variations were shown to take place on the order of days, in some cases modified by dust concentration (Lavaysse et al. 2011), while its southward expansion on sub-seasonal timescale has been associated with higher shear and more intense MCSs in SWA (e.g. Talib et al. 2022)."

**Other specific comments/questions**
L135: "TCWV represents the precipitable water the atmosphere holds better than the humidity." I do not get this sentence.

This statement has been removed since the preceding sentence already covers the point made.

L140: "km2". Set the "2" in superscript.

The change has been effected

L182: "Based on 6 different large-scale node patterns …" Should be nine!?

Thank you for catching this. Changed!

Fig. 3: Colours + patterns for the within-season nodes are visually not necessarily well distinguishable. It helps though that the bars are ordered the same way as in the legend.

Thank you for this comment. We changed the pattern colour for this plot for better visibility with the image below:

[Figure]

Fig. 8: It appears to me that signals in node 1 are dominated by land-sea breeze convection along the coast which are gradually suppressed in nodes 4 and 7. Therefore, the large-scale settings seemingly facilitate such rather local-scale developments. Maybe the authors can briefly pick up on this in the text.

Thank you for this suggestion, it has been considered in the text in ll. X as follows:
"Frequency signals in node 1 are dominated by land-sea breeze convection along the coast which are gradually suppressed in nodes 4 and 7. Large-scale settings, therefore, seemingly facilitate such rather local-scale developments. "

L359: "… reveal a widespread increase in zonal wind shear anomaly…". But in this case, it means (mostly) a weaker westerly shear? The authors may work with directional indications for clarity.

Thank you for catching this. Directional indications have been made to the section on zonal wind shear for clarity

**Referee #2**
We thank the referee for their valuable input, which has helped to improve the clarity of our manuscript and figures. The line references below refer to the newly changed manuscript. Additionally, we provide a document with tracked changes.

**General comments**:
I find the revised manuscript as a significant improvement. The full SOM appears to better grasp the variability in the system, as is evident by the sharp clustering in Fig. 13. The persistency-transition analysis and the spatial variability of MCSs greatly enhance the scope of the paper, and lead to valuable findings while enabling a deeper understanding of the SOM analysis.

I think the manuscript should be accepted for publication following several minor comments:

Thank you very much for such encouraging comments.

Minor comments:
L1: Be consistent with capital letters.

Changes have been implemented to this effect

L21: I suggest adding a motivational sentence stating the importance of the topic.

Thank you for this comment. The sentence "These MCS events are the dominating rain-bearing systems, contributing over 50% of annual rainfall over SWA." has been added to the abstract stating the importance of the topic.

L35: … the SOM identified…

This has been corrected

L145: The sentence is incomplete.

The sentence has been completed to read: "Here, only land-based MCSs are considered because MCSs over land are fundamentally more intense and deep than its counterpart over the ocean (Mohr and Zipser 1996)."

L174: analysis

This change has been effected

L175: remove "to choose"

This has been removed

L316: MCSs rarely develop…

Thank you for pointing this out. Changed!

L317: address the similar frequencies of clusters 1 and 9.

We now added the following sentence: "Nodes 1 and 9 feature the same overall MCS frequency, where node 1 however shows coastal MCS frequency peaks as is representative for dry season characteristics, while MCS frequency peaks are shifted towards the Sahel during node 9 monsoon conditions."

L383: I would replace "swath" with a more common word.

The word "swath" has been replaced with "broad strip". The statement now reads: "During monsoon nodes, node 3 shows a broad strip of high CAPE values in particular to the coast and in some instances extends to the entire SWA (node 6) and north of SWA (node 9)."

L439: arrange the spacing.

This has been corrected

L469: southward

This has been corrected

L471: why "presumably"? Is this not shown in Fig. 9?

Thank you for pointing this out. This is well shown in Fig. 9. We have, therefore, removed the word 'presumably' from the statement.

L491-492: Why then don't these nodes show maximum MCS frequencies?

Thank you for raising this point. Note that the climatologies as presented in Fig. 13 represent conditions at the locations where MCSs were sampled rather than a full-domain mean. This means that, while monsoon nodes may have higher TCWV and similar shear as compared to transition nodes, the domain area affected by these favourable MCS conditions may still be smaller during monsoon months, which thus feature a lower MCS frequency. For transition nodes, a larger domain area is indeed affected by favourable MCS conditions (c.f. Fig 5, Fig 6, with moisture and shear maxima covering most of SWA for the transition nodes, while only limited areas are affected for monsoon months with maxima shifted to the north, outside of the domain). Hence, most of the MCSs and rainfall over southern West Africa occur during the transition season. This is now clarified in the concluding statements under Section 4.3 to read: "Note that while monsoon months feature higher TCWV and similar shear conditions compared to transition nodes for MCS-location climatologies in Fig. 13, a larger domain area is affected by MCS-favourable conditions for transition nodes (c.f. Figs. 5,6). As a consequence, transition nodes exhibit higher overall MCS frequencies."